# Myotubularin related protein-2 and its phospholipid substrate PIP$_2$ control Piezo2-mediated mechanotransduction in peripheral sensory neurons

Pratibha Narayanan[1], Meike Hütte[1], Galina Kudryasheva[2], Francisco J Taberner[3], Stefan G Lechner[3], Florian Rehfeldt[2], David Gomez-Varela[1], Manuela Schmidt[1]*

[1]Emmy Noether-Group Somatosensory Signaling and Systems Biology, Max Planck Institute for Experimental Medicine, Goettingen, Germany; [2]Third Institute of Physics - Biophysics, University of Goettingen, Goettingen, Germany; [3]Institute of Pharmacology, Heidelberg, Germany

**Abstract** Piezo2 ion channels are critical determinants of the sense of light touch in vertebrates. Yet, their regulation is only incompletely understood. We recently identified myotubularin related protein-2 (Mtmr2), a phosphoinositide (PI) phosphatase, in the native Piezo2 interactome of murine dorsal root ganglia (DRG). Here, we demonstrate that Mtmr2 attenuates Piezo2-mediated rapidly adapting mechanically activated (RA-MA) currents. Interestingly, heterologous Piezo1 and other known MA current subtypes in DRG appeared largely unaffected by Mtmr2. Experiments with catalytically inactive Mtmr2, pharmacological blockers of PI(3,5)P$_2$ synthesis, and osmotic stress suggest that Mtmr2-dependent Piezo2 inhibition involves depletion of PI(3,5)P$_2$. Further, we identified a PI(3,5)P$_2$ binding region in Piezo2, but not Piezo1, that confers sensitivity to Mtmr2 as indicated by functional analysis of a domain-swapped Piezo2 mutant. Altogether, our results propose local PI(3,5)P$_2$ modulation via Mtmr2 in the vicinity of Piezo2 as a novel mechanism to dynamically control Piezo2-dependent mechanotransduction in peripheral sensory neurons.
DOI: https://doi.org/10.7554/eLife.32346.001

*For correspondence:
mschmidt@em.mpg.de

**Competing interests:** The authors declare that no competing interests exist.

## Introduction

Our sense of touch relies on mechanotransduction, that is the conversion of mechanical stimuli to electrical signals in primary afferent sensory neurons of the somatosensory system. Piezo2 ion channels have emerged as major somatosensory mechanotransducers and mediate rapidly adapting mechanically activated (RA-MA) currents in sensory neurons, such as those of dorsal root ganglia (DRG) (*Coste et al., 2010*, *2012*). By now it has been established that Piezo2 is crucially involved in vertebrate light touch and proprioception (*Coste et al., 2010*, *2012*; *Florez-Paz et al., 2016*; *Ranade et al., 2014*; *Woo et al., 2015*; *Woo et al., 2014*). Despite its importance, the regulation of native Piezo2 is only beginning to be elucidated. Mechanistically, several scenarios might be at play (*Wu et al., 2017*): direct action of phospholipids, the modulation of local membrane properties and protein-protein interactions, just to name a few. Studies on other mechanosensitive ion channels such as the family of small conductance channels (MscS) (*Sukharev, 2002*) as well as eukaryotic TRAAK, TREK1 (*Brohawn et al., 2014a*, *2014b*) and Piezo1 (*Cox et al., 2016*; *Lewis and Grandl, 2015*; *Wu et al., 2016*) demonstrated their direct interplay with components of the lipid bilayer. In case of Piezo2, RA-MA currents were shown to be inhibited through depletion of phosphatidylinositol 4,5-bisphosphate (PI(4,5)P$_2$) and phosphatidylinositol 4-monophosphate (PI(4)P) (*Borbiro et al., 2015*), and also by depletion of cholesterol at the plasma membrane (*Qi et al., 2015*). Besides, lipid-

**eLife digest** We often take our sense of touch for granted. Yet, our every-day life greatly depends on the ability to perceive our environment to alert us of danger or to further social interactions, such as mother-child bonding. Our sense of touch relies on the conversion of mechanical stimuli to electrical signals (this is known as mechanotransduction), which then travel to brain to be processed. This task is fulfilled by specific ion channels called Piezo2, which are activated when cells are exposed to pressure and other mechanical forces. These channels can be found in sensory nerves and specialized structures in the skin, where they help to detect physical contact, roughness of surfaces and the position of our body parts.

It is still not clear how Piezo2 channels are regulated but previous research by several laboratories suggests that they work in conjunction with other proteins. One of these proteins is the myotubularin related protein-2, or Mtmr2 for short. Now, Narayanan et al. – including some of the researchers involved in the previous research – set out to advance our understanding of the molecular basis of touch and looked more closely at Mtmr2.

To test if Mtmr2 played a role in mechanotransduction, Narayanan et al. both increased and reduced the levels of this protein in sensory neurons of mice grown in the laboratory. When Mtmr2 levels were low, the activity of Piezo2 channels increased. However, when the protein levels were high, Piezo2 channels were inhibited. These results suggest that Mtmr2 can control the activity of Piezo2. Further experiments, in which Mtmr2 was genetically modified or sensory neurons were treated with chemicals, revealed that Mtmr2 reduces a specific fatty acid in the membrane of nerve cells, which in turn attenuates the activity of Piezo2.

This study identified Mtmr2 and distinct fatty acids in the cell membrane as new components of the complex setup required for the sense of touch. A next step will be to test if these molecules also influence the activity of Piezo2 when the skin has become injured or upon inflammation.

DOI: https://doi.org/10.7554/eLife.32346.002

or cytoskeleton-induced changes in plasma membrane tension have been shown to impact somatosensory mechanotransduction (*Jia et al., 2016*; *Morley et al., 2016*; *Qi et al., 2015*). In contrast to the vast knowledge of the molecular network governing mechanotransduction in the nematode C. elegans, to date only few protein-protein interactions relevant for Piezo2 physiology have been identified in vertebrates. These include stomatin-like protein STOML3 (*Poole et al., 2014*; *Qi et al., 2015*; *Wetzel et al., 2007*), unidentified protein tethers to the extracellular matrix (*Hu et al., 2010*) and Pericentrin (*Narayanan et al., 2016*). It is noteworthy that all of these exhibit pronounced effects on the magnitude or mechanical sensitivity of Piezo2 RA-MA currents.

In order to advance the molecular understanding of Piezo2 regulation, we recently performed an interactomics screen, which revealed several additional binding partners of native Piezo2 in murine DRG (*Narayanan et al., 2016*). A significantly enriched and prominent member of the Piezo2 interactome was myotubularin related protein 2 (Mtmr2) (*Narayanan et al., 2016*), a phosphoinositide phosphatase of the MTMR family (*Bolino et al., 2002*). Interestingly, Mtmr2 was previously described to be highly expressed in DRG sensory neurons and Schwann cells (*Bolino et al., 2002*). Functionally, Mtmr2 catalyzes the removal of a 3-phosphate group from its phosphoinositide (PIPs) substrates phosphatidylinositol 3-monophosphate PI(3)P and phosphatidylinositol 3,5-bisphosphate PI(3,5)$P_2$ (*Begley et al., 2006*; *Berger et al., 2002*). Remarkably, PI(3,5)$P_2$ is much less abundant than most other PIPs, for example PI(4,5)$P_2$ (*Zolov et al., 2012*), but can be rapidly and transiently regulated by a large enzymatic protein complex in response to cellular stimulation (*McCartney et al., 2014a*; *Vaccari et al., 2011*; *Zolov et al., 2012*). Hence, PI(3,5)$P_2$ is exquisitely suited to control rapid cellular signaling events (*Ho et al., 2012*; *Ikonomov et al., 2007*; *Li et al., 2013a*; *McCartney et al., 2014a*; *Zhang et al., 2012*), and also the activity of receptors and ion channels (*Dong et al., 2010*; *Ho et al., 2012*; *Klaus et al., 2009*; *McCartney et al., 2014a*; *McCartney et al., 2014b*; *Tsuruta et al., 2009*). Altogether, this raises the question whether Mtmr2 and its PIP substrates are implicated in Piezo2-mediated mechanotransduction and generally, in somatosensory mechanosensation.

Here, we show that Mtmr2 levels control Piezo2-mediated RA-MA currents. While elevated Mtmr2 expression attenuated Piezo2 RA-MA currents, siRNA-mediated knockdown of Mtmr2 resulted in Piezo2 RA-MA current potentiation. Interestingly, heterologously expressed Piezo1 and other known subtypes of MA currents in DRG were largely unaffected. Mechanistically, our experiments with catalytically inactive Mtmr2, pharmacological inhibitors, and osmotic stress suggest that changes in the levels of PI(3,5)P$_2$ regulate Piezo2 RA-MA currents. In line with these findings we uncovered a previously unknown polybasic motif in Piezo2 that can bind PI(3,5)P$_2$ and confers Piezo2 sensitivity to Mtmr2. Collectively, our study reveals a link between Mtmr2 activity and PI(3,5)P$_2$ availability to locally control Piezo2 function.

## Results

### Mtmr2 suppresses Piezo2-mediated RA-MA currents

Our previous work revealed Mtmr2 as a highly enriched member of the native Piezo2 interactome in DRG (significance of identification: p=0.00030, unpaired t-test; enrichment factor: log$_2$ 9.96) (*Narayanan et al., 2016*). We first validated the reported expression of Mtmr2 in peripheral sensory neurons of DRG (*Previtali et al., 2003*; *Vaccari et al., 2011*) including those neurons expressing Piezo2 (*Figure 1—figure supplement 1a,b*). For a more detailed subcellular analysis we used the proximity ligation assay (PLA). In this way we could show the close vicinity of Piezo2 and Mtmr2 in both, somata and neurites of cultured DRG neurons, and upon co-transfection in HEK293 cells (*Figure 1a–d* and *Figure 1—figure supplement 1c,d*). It is important to note here that the PLA technique is prone to high background upon heterologous expression as shown by our additional control experiments in HEK293 cells (*Figure 1—figure supplement 1d*). In these, we co-overexpressed Piezo2 with TRPA1 and Vti1b, respectively. Both of these controls exhibited clear PLA signal (likely attributable to massive overexpression upon transfection), though less than co-overexpression with Mtmr2 (*Figure 1—figure supplement 1d*).

Following, we wanted to assess whether Mtmr2 affects Piezo2 function. To this end we performed electrophysiological measurements of Piezo2-mediated RA-MA currents in HEK293 cells, which represent a well-defined heterologous system to study Piezo2 function (*Coste et al., 2010*; *Poole et al., 2014*). Interestingly, Mtmr2 co-expression led to a pronounced reduction of Piezo2-mediated RA-MA currents compared to mock conditions (*Figure 1e,f*). Moreover, the displacement threshold of RA-MA currents was significantly increased upon Mtmr2 overexpression indicating the requirement of stronger mechanical stimulation to reach a threshold current amplitude (*Supplementary file 1*; please see Materials and methods for details on the calculation of the displacement threshold) (*Eijkelkamp et al., 2013*; *Morley et al., 2016*; *Narayanan et al., 2016*). Importantly, the inactivation time constant of Piezo2 currents remained unchanged upon Mtmr2 co-expression (*Supplementary file 1*). These data indicate that Mtmr2 overexpression suppresses Piezo2 currents in HEK293 cells while maintaining their defining property, that is rapid adaptation. To exclude that Mtmr2 overexpression renders cells unhealthy and might therefore unspecifically suppress Piezo2 currents, we co-expressed Mtmr2 with Kv1.1 and measured voltage-gated currents, which were similar to mock transfected controls (*Figure 1—figure supplement 2a*). In addition, we asked whether Mtmr2 may modulate MA currents generated by Piezo1, the homologue of Piezo2 (*Coste et al., 2010*). Remarkably, we did not observe any differences in Piezo1 currents upon overexpression of Mtmr2 (*Figure 1—figure supplement 2b* and *Supplementary file 1*). These results suggest a certain degree of specificity of the functional Piezo2-Mtmr2 interaction.

Next, we aimed at assessing whether Mtmr2 can modulate native Piezo2 RA-MA currents in cultured DRG neurons, as well. DRG cultures allow for the targeted manipulation of protein levels by nucleofection and concomitant assessment of Piezo2-mediated RA-MA currents (*Coste et al., 2010*; *Narayanan et al., 2016*). It is also noteworthy, that the DRG culture system has been employed for the original discovery (*Coste et al., 2010*) and further characterization of Piezo2 RA-MA currents (*Dubin et al., 2012*; *Eijkelkamp et al., 2013*; *Jia et al., 2016*; *Narayanan et al., 2016*; *Poole et al., 2014*; *Qi et al., 2015*); hence it serves as a model for sensory transduction processes (*Coste et al., 2007*; *Coste et al., 2010*; *Lechner and Lewin, 2009*).

In order to test whether the effect on Piezo2 RA-MA currents observed in HEK293 cells can be recapitulated, we overexpressed Mtmr2 in DRG cultures and measured RA-MA currents. Similar to

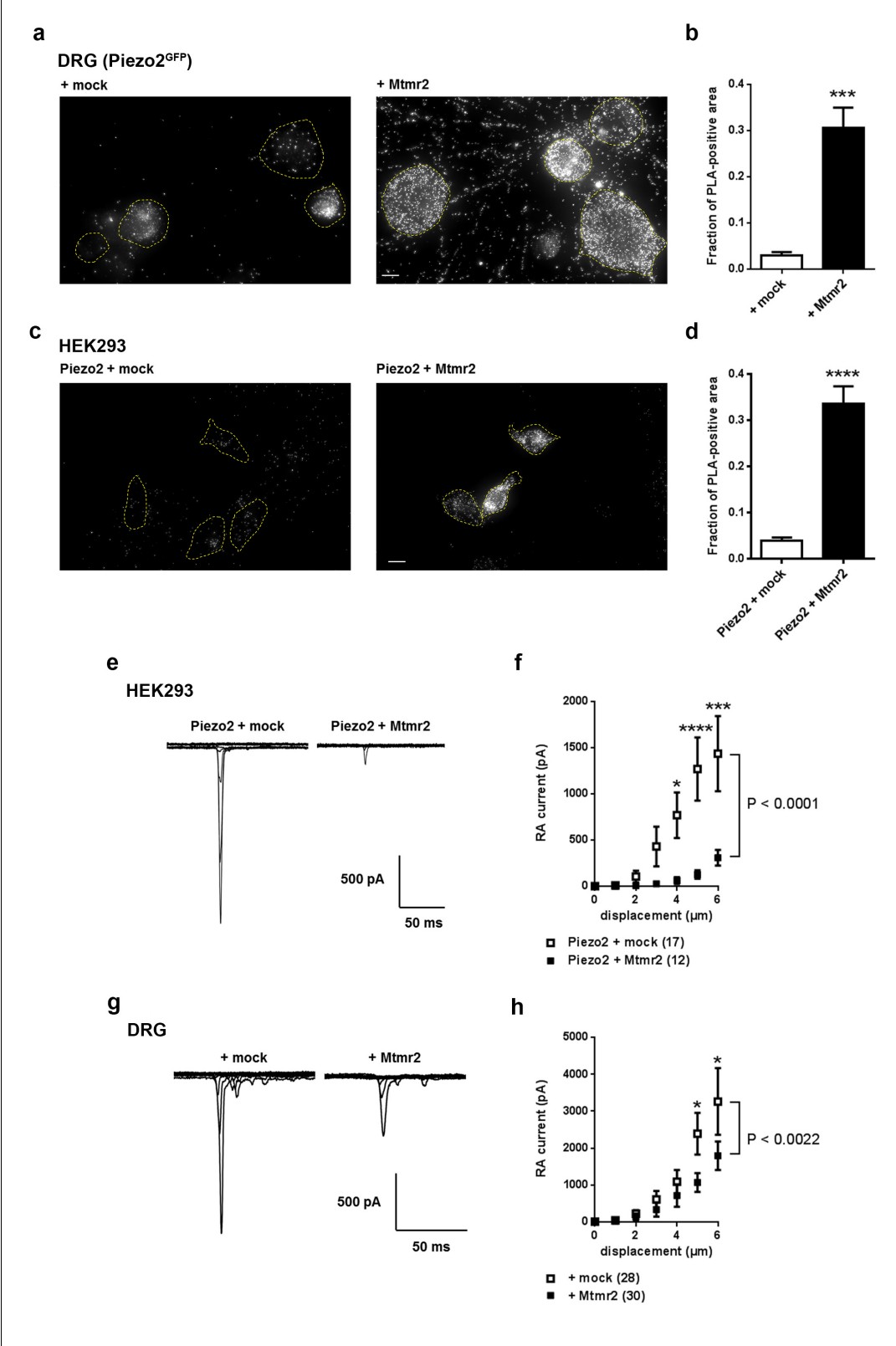

**Figure 1.** Mtmr2 suppresses Piezo2-mediated RA-MA currents in HEK293 cells and DRG neurons. (**a–d**) Representative images (**a,c**) and quantification (**b,d**) of a proximity ligation assay (PLA) in cultured DRG neurons (**a,b**) of *Piezo2^GFP* mice (*Woo et al., 2014*) and HEK293 cells (**c,d**). As anti-Mtmr2 antibodies failed to work in neuronal cultures, DRG were transfected with Mtmr2-myc or mock-myc and PLA was performed with antibodies against Piezo2 and myc. Please note the distribution of the PLA signal in soma and neurites of DRG. HEK293 cells were co-transfected with Piezo2-GST-IRES-
*Figure 1 continued on next page*

*Figure 1 continued*

GFP and Mtmr2-myc or Piezo2-GST-IRES-GFP and mock-myc and PLA was performed with antibodies against GST and myc. Only cells with pronounced GFP signal (due to expression of pmaxGFPVector in DRG and Piezo2-GST-IRES-GFP in HEK293 cells) were considered for the analysis. Cell boundaries are demarcated in yellow. In both cell types, DRG and HEK293 cells, transfection of Mtmr2-myc exhibited significantly stronger PLA signal compared to controls (b,d). Scale bar: 10 µm. Quantification of the total area of PLA signal/total soma area (fraction of PLA-positive area) in DRG cultures (p<0.0001; Mann-Whitney test; + mock: n = 53 neurons; + Mtmr2-myc: n = 53 neurons) (b). The quantification of the intensity of PLA signal in neurites of cultured DRG neurons can be found in *Figure 1—figure supplement 1c*. Quantification of the total area of PLA signal/total cell area in HEK293 cells (fraction of PLA-positive area) (p<0.0001; Mann-Whitney test; Piezo2-GST + mock: n = 60 cells; Piezo2-GST + Mtmr2-myc: n = 54 cells) (d). Additional controls for PLA in HEK293 cells can be found in *Figure 1—figure supplement 1d*. (e) Representative traces of RA-MA currents in HEK293 cells upon co-expression of Piezo2 with mock or Mtmr2 and (f) stimulus-current curves. Overexpression of Mtmr2 suppressed Piezo2 current magnitudes compared to mock overexpression (Piezo2 + mock: n = 17 cells; Piezo2 + Mtmr2: n = 12 cells; 2-way ANOVA suggested a significant effect (P<0.0001) of Mtmr2 overexpression on Piezo2 currents; Holm-Sidak's multiple comparisons test was used to compare both conditions at individual stimulus magnitudes, p-values are indicated by * in the graph). The displacement threshold was increased upon co-expression of Mtmr2 (p=0.0098; Mann-Whitney test; *Supplementary file 1*). The inactivation time constant of RA-MA currents remained unchanged (*Supplementary file 1*). (g) Representative traces of RA-MA currents in primary cultures of DRG neurons and (h) stimulus-current curves showed a significant decrease in RA-MA current magnitude upon overexpression of Mtmr2 compared to mock ( + mock: n = 28 neurons; + Mtmr2: n = 30 neurons; 2-way ANOVA suggested a significant effect (P<0.0022) of Mtmr2 overexpression on RA-MA currents; Holm-Sidak's multiple comparison test was performed to compare both conditions at individual stimulus magnitudes, p-values are indicated by * in the graph). The displacement threshold and inactivation time constant of RA-MA currents were not changed upon overexpression of Mtmr2 in DRG neurons (*Supplementary file 1*).

DOI: https://doi.org/10.7554/eLife.32346.003

The following figure supplements are available for figure 1:

**Figure supplement 1.** Mtmr2 is expressed in mouse DRG and also in close vicinity to Piezo2.
DOI: https://doi.org/10.7554/eLife.32346.004

**Figure supplement 2.** Mtmr2 overexpression does not influence Kv1.1- or Piezo1-mediated currents.
DOI: https://doi.org/10.7554/eLife.32346.005

HEK293 cells, Mtmr2 overexpression suppressed native RA-MA currents in DRG cultures when compared to mock transfection (*Figure 1g,h*). The displacement threshold was unaffected upon Mtmr2 overexpression (*Supplementary file 1*), which is in contrast to our results in HEK293 cells potentially reflecting differences in Piezo2/Mtmr2 stoichiometry or the contribution of unknown neuronal modulators. Also, the inactivation time constant of RA-MA currents was unchanged upon overexpression of Mtmr2 in DRG neurons (*Supplementary file 1*).

## Mtmr2 knockdown potentiates Piezo2 activity in peripheral sensory neurons

We went on to investigate whether Mtmr2 downregulation could potentiate native RA-MA currents in DRG. Successful knockdown of *Mtmr2* was achieved after 72 hr and evaluated by quantitative PCR (*Figure 2—figure supplement 1a*). We assessed *Piezo2* mRNA levels, Piezo2 membrane expression and the percentage of Piezo2-positive neurons and did not observe any changes upon Mtmr2 knockdown (*Figure 2—figure supplement 1b–d*). However, when we measured RA-MA currents in *Mtmr2* siRNA-nucleofected DRG cultures, we observed a significant augmentation in current amplitude whereas the displacement threshold and inactivation time constant were unchanged (*Figure 2a–b*; *Supplementary file 1*). These results are in agreement with our data on Mtmr2 overexpression (*Figure 1*) and suggest that Mtmr2 levels can modulate Piezo2 function: decreased expression of Mtmr2 potentiated, while increased expression suppressed Piezo2 RA-MA currents.

Besides Piezo2-mediated RA-MA currents, cultured DRG neurons display two other major types of MA currents: intermediately- (IA) and slowly-adapting- (SA) MA currents categorized based on their inactivation time constant (please see Materials and methods for details) (*Coste et al., 2007*; *Coste et al., 2010*; *Hu and Lewin, 2006*; *Lechner and Lewin, 2009*). These seemed to be largely unaltered by Mtmr2 knockdown (*Figure 2c,d*; *Supplementary file 1*) indicating that mechanotransduction is not generally compromised. Remarkably though, upon Mtmr2 knockdown we detected a significant redistribution of the number of cells exhibiting MA subpopulations: a moderate increase in the proportion of RA-MA cells paralleled by a decrease of the IA-MA population (*Figure 2e*). To date the interpretation of the latter is difficult and requires yet to be obtained insights into the molecular nature of IA-MA currents.

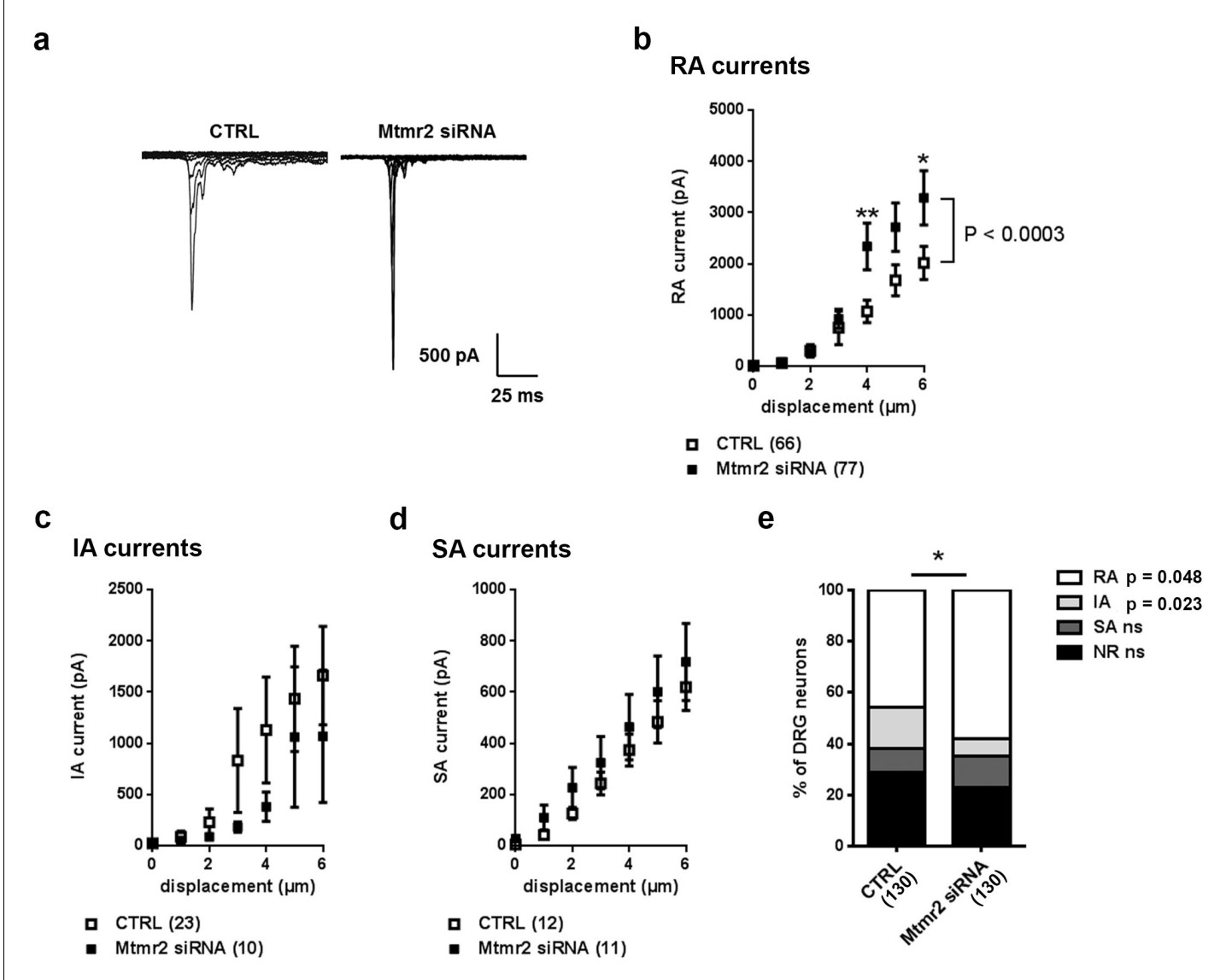

**Figure 2.** Mtmr2 knockdown potentiates Piezo2 activity in peripheral sensory neurons. (**a**) Representative traces of RA-MA currents in primary cultures of DRG neurons and (**b**) stimulus-current curves for RA-MA currents upon nucleofection of *Mtmr2* siRNA showed a significant increase in RA-MA current magnitude compared to nucleofection with AllStar Negative Control siRNA (CTRL: n = 66 neurons; *Mtmr2* siRNA: n = 77 neurons; 2-way ANOVA suggested a pronounced effect (P<0.0003) of Mtmr2 knockdown on RA-MA currents; Holm-Sidak's multiple comparisons test was used to compare both conditions at individual stimulus magnitudes, p-values are indicated by * in the graph). The displacement threshold and inactivation time constant of RA-MA currents remained unchanged upon knockdown of Mtmr2 (*Supplementary file 1*). Of note, MA current properties cannot be compared between different experiments or treatments of DRG cultures, for example *Figure 2b* cannot be compared to overexpression of Mtmr2 (*Figure 1*). Cultures were differently nucleofected (siRNA vs. plasmids) and recorded on different days in vitro (DIV) according to established protocols (please see Materials and methods for details). Hence matching controls were performed for each set of data. (**c**) Stimulus-current curves show IA-MA currents were unaffected by knockdown of Mtmr2 in DRG neurons (CTRL: n = 23 neurons; *Mtmr2* siRNA: n = 10 neurons; ns; 2-way ANOVA). (**d**) Stimulus-current curves show SA-MA currents were unchanged upon Mtmr2 knockdown (n = 11–12 neurons per condition; ns; 2-way ANOVA). Of note, the displacement thresholds and inactivation time constants of IA-MA and SA-MA currents remained unchanged upon *Mtmr2* siRNA nucleofection (*Supplementary file 1*). (**e**) Stacked histograms show the number of cells exhibiting different MA currents upon knockdown of Mtmr2 in cultured DRG. The proportions of cells exhibiting RA and IA currents were significantly changed in cultures transfected with *Mtmr2* siRNA. RA:IA:SA:NR (% of total; rounded to whole numbers): CTRL: 46:16:9:29; *Mtmr2* siRNA: 58:7:12:23; p<0.044 overall and for the proportion of RA/total (p=0.048) and IA/total (p=0.023), respectively; Chi-square test;≥130 neurons were analyzed per condition). NR (non-responsive), refers to cells which showed no MA current.

DOI: https://doi.org/10.7554/eLife.32346.006

The following figure supplement is available for figure 2:

**Figure supplement 1.** Mtmr2 knockdown in cultured DRG neither affects Piezo2 mRNA nor Piezo2 membrane levels or overall expression.

*Figure 2 continued on next page*

*Figure 2 continued*

DOI: https://doi.org/10.7554/eLife.32346.007

## Mtmr2 modulates Piezo2-mediated mechanotransduction largely via PI(3,5)P$_2$

Next we asked how, on a mechanistic level, Mtmr2 could regulate Piezo2 activity given that neither Piezo2 mRNA levels nor membrane expression seemed to be affected (*Figure 2—figure supplement 1b,c*). Mtmr2 is a phosphatase that catalyzes the removal of the 3-phosphate group from PI(3)P as well as PI(3,5)P$_2$ (*Berger et al., 2002*). Under resting conditions PI(3,5)P$_2$ is present at low levels (*Jin et al., 2016*; *McCartney et al., 2014a*) but is transiently and steeply generated upon a diverse range of cellular stressors. This is in contrast to intensely studied and highly abundant PI(4,5)P$_2$ known to regulate ion channels (*Gamper and Shapiro, 2007*) including both, Piezo1 and Piezo2 (*Borbiro et al., 2015*).

To explore a potential role of Mtmr2 phosphatase activity and the resulting change in PIP levels (*Laporte et al., 1998*; *McCartney et al., 2014b*; *Mironova et al., 2016*; *Previtali et al., 2007*) for the regulation of Piezo2 currents, we generated a catalytically inactive Mtmr2 mutant (Mtmr2C417S) by substituting Cysteine 417 for Serine (*Berger et al., 2002*). If the catalysis of PIPs was essential to the functional interaction of Piezo2-Mtmr2, the catalytically inactive mutant should fail to suppress Piezo2 currents when overexpressed in HEK293 cells (please see our data on wild type Mtmr2 in *Figure 1e,f*). This was indeed found to be the case. Mtmr2C417S co-expression did not suppress Piezo2-mediated currents in HEK293 cells as determined by comparison of stimulus-current curves (*Figure 3a*) with mock transfected cells. Instead, we observed a trend towards moderate potentiation of Piezo2 currents upon co-expression of Mtmr2C417S (especially at low stimulus magnitudes; *Figure 3a*). This result strongly suggests that the catalytic activity of Mtmr2 and the consequential alteration of PIP levels (*Laporte et al., 1998*; *McCartney et al., 2014b*; *Mironova et al., 2016*;

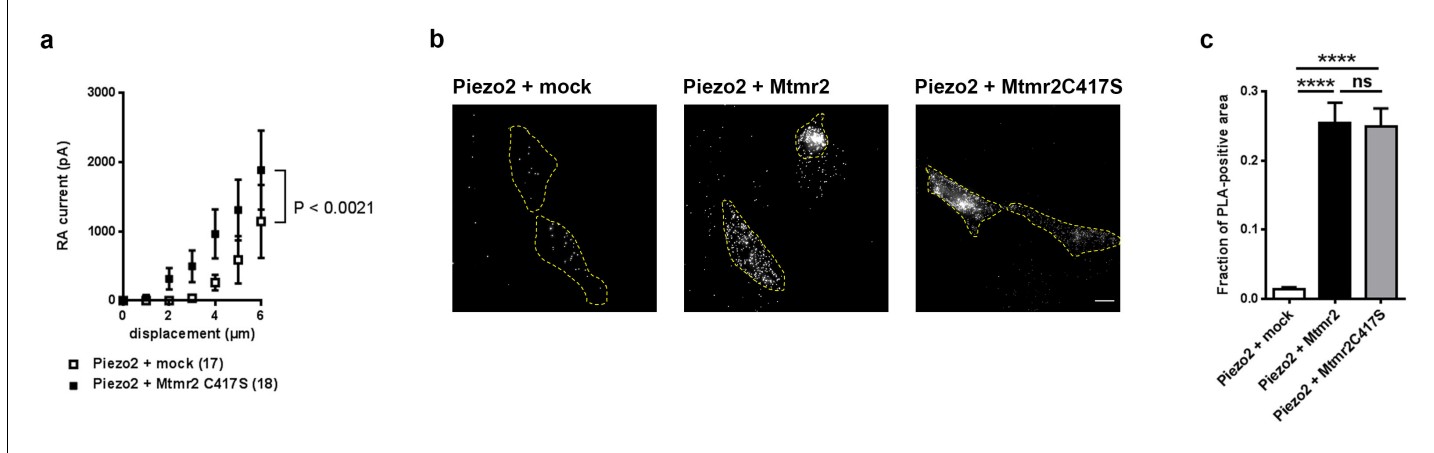

**Figure 3.** Catalytic activity of Mtmr2 is necessary to suppress Piezo2-mediated RA-MA currents. (a) Stimulus-current curves upon co-expression of Piezo2 with the catalytically inactive Mtmr2 C417S mutant in HEK293 cells compared to mock controls. Mtmr2 C417S overexpression slightly, but not significantly increased Piezo2 RA-MA currents especially at lower stimulus magnitudes (Piezo2 + mock: n = 17 cells; Piezo2 + Mtmr2 C417S: n = 18 cells). 2-way ANOVA reported a significant (P<0.0021) overall effect of Mtmr2 C417S overexpression on RA-MA currents, however a Holm-Sidak's multiple comparisons test showed no significant difference between currents at individual stimulus magnitudes.). Of note, the displacement thresholds and inactivation time constant were unaffected upon overexpression with Mtmr2 C417S compared to mock (*Supplementary file 1*). (b–c) Representative images (b) and quantification (c) of PLA signal using antibodies against GST and myc to detect Piezo2-GST-IRES-GFP and Mtmr2 C417S-myc, respectively. PLA signal upon co-transfection of Piezo2-GST + Mtmr2 C417S-myc was indistinguishable from Piezo2-GST + Mtmr2 myc and significantly stronger than Piezo2-GST + mock. Cell boundaries are demarcated in yellow. Only cells with pronounced GFP signal (due to expression of Piezo2-GST-IRES-GFP) were considered for the analysis. Scale bar: 10 µm. Quantification of the number of the total area of PLA signal/total cell area (fraction of PLA-positive area) (c); p<0.0001 compared to Piezo2-GST + mock, Kruskal-Wallis test followed by Dunn's Multiple Comparison Test; Piezo2-GST + mock: n = 75 cells; Piezo2-GST + Mtmr2-myc: n = 70 cells; Piezo2-GST + Mtmr2 C417S-myc: n = 70 cells.

DOI: https://doi.org/10.7554/eLife.32346.008

*Previtali et al., 2007*) may underlie the regulation of Piezo2 by Mtmr2. Importantly, enzymatically inactive Mtmr2C417S was abundantly expressed and remained capable of associating with Piezo2 in close proximity as suggested by PLA upon co-overexpression of Piezo2 and Mtmr2C417S in comparison to wild type Mtmr2 (*Figure 3b–c*).

Based on the results obtained from experiments with Mtmr2C417S we then proceeded to manipulate neuronal PIP levels. *Figure 4a* illustrates a schematic view of the PIP synthesis and turnover pathway Mtmr2 is involved in and also indicates pharmacological inhibitors known to intervene with this pathway (*Vaccari et al., 2011*). According to this scheme and a wealth of previous work (*Laporte et al., 1998*; *McCartney et al., 2014b*; *Mironova et al., 2016*; *Previtali et al., 2007*; *Vaccari et al., 2011*), knockdown of Mtmr2 would increase the levels of PI(3)P and even more PI(3,5)$P_2$ (*Vaccari et al., 2011*). We tried to experimentally mimic heightened levels of these PIPs by inclusion of exogenous PIPs in the intracellular recording solution (*Dong et al., 2010*); however, we did not see any change in RA-MA currents (*Figure 4—figure supplement 1a*). Due to technical factors that may confound this data (e.g. rapid breakdown of exogenous PIPs by intracellular phosphatases), we opted to perform additional experiments. We reversed the described accumulation of these two PIPs upon Mtmr2 knockdown (*Laporte et al., 1998*; *McCartney et al., 2014b*; *Mironova et al., 2016*; *Previtali et al., 2007*; *Vaccari et al., 2011*) by applying commonly-used inhibitors of PIP synthesis, that is Wortmannin, an inhibitor of the class III PI 3-kinase, (*Messenger et al., 2015*), and Apilimod, an inhibitor of PIKfyve (*Cai et al., 2013*; *Vaccari et al., 2011*), respectively (please see scheme in *Figure 4a*). If elevated levels of PI(3)P or PI(3,5)$P_2$ were implicated in Mtmr2-knockdown-induced potentiation of Piezo2, the presence of the corresponding inhibitor in the recording solution would be expected to counteract this potentiation. Wortmannin application only marginally altered the increase in RA-MA currents upon Mtmr2 knockdown in neurons (*Figure 4b*). Apilimod, on the other hand, significantly diminished the magnitude of RA-MA currents in *Mtmr2* siRNA-treated neurons (*Figure 4b*). In line with a reversal of Mtmr2-induced potentiation, the displacement threshold of RA-MA currents was significantly increased upon Apilimod treatment (*Supplementary file 1*). Interestingly, Apilimod treatment in wild type DRG neurons did not affect RA-MA currents (*Figure 4—figure supplement 1b*; *Supplementary file 1*), as expected given the low cellular expression and tight regulation of PI(3,5)$P_2$ under physiological conditions (*Jin et al., 2016*; *McCartney et al., 2014a*). Taken together, our results point towards a role of PI(3,5)$P_2$ for the functional interaction of Piezo2 and Mtmr2.

We then intended to assess the significance of PI(3,5)$P_2$ availability for Piezo2 function in a more physiological setting. In peripheral sensory neurons changes in cellular osmolarity cause activation of diverse ion channels and receptors followed by initiation of various signaling pathways involved in volume regulation (*Lechner et al., 2011*; *Liu et al., 2007*; *Quallo et al., 2015*). Interestingly, previous work has indicated that also Mtmr2, PI(3,5)$P_2$ (*Berger et al., 2003*; *Dove et al., 1997*) and Piezo2 (*Jia et al., 2016*) can be modulated by osmotic stress. Upon hypoosmotic stress Mtmr2 trafficking was altered and PI(3,5)$P_2$ levels were reported to be elevated in eukaryotic cells (*Berger et al., 2003*; *Dove et al., 1997*). In the case of Piezo2, hypoosmotic stress was shown to potentiate Piezo2 RA-MA currents, which was independent of the prominent osmosensor TRPV4 (*Jia et al., 2016*). Therefore, we employed osmotic stress as a physiological stimulus to investigate the link between Mtmr2, PI(3,5)$P_2$ and Piezo2. First, we confirmed the previously described (*Jia et al., 2016*) potentiation of Piezo2 RA-MA currents by application of extracellular hypotonic stress to DRG cultures (*Figure 4—figure supplement 1c*). We then postulated that an increase of PI(3,5)$P_2$ levels by extracellular hypotonicity (*Berger et al., 2003*; *Dove et al., 1997*) should counteract the RA-MA suppression upon Mtmr2-overexpression, which we described in *Figure 1h* above. Indeed, in Mtmr2 overexpressing DRG cultures we recorded significantly higher RA-MA currents under extracellular hypotonic conditions compared to isotonic conditions (*Figure 4c*). Other RA-MA current parameters were unchanged (*Supplementary file 1*). We then tested whether the opposite was also true: Would an expected decrease of PI(3,5)$P_2$ levels by intracellular hypotonicity (*Dove et al., 1997*) prevent RA-MA current sensitization upon Mtmr2 knockdown? Conceptually, this experiment is analogous to Apilimod application in *Figure 4b* above, where we pharmacologically inhibited PI(3,5)$P_2$ production in DRG cultures. As predicted, under intracellular hypotonic conditions RA-MA currents were significantly smaller in siRNA-treated cultures compared to isotonic conditions (*Figure 4d*; *Supplementary file 1*). In parallel, the displacement threshold was significantly increased, while other RA-MA current parameters remained unchanged (*Supplementary file*

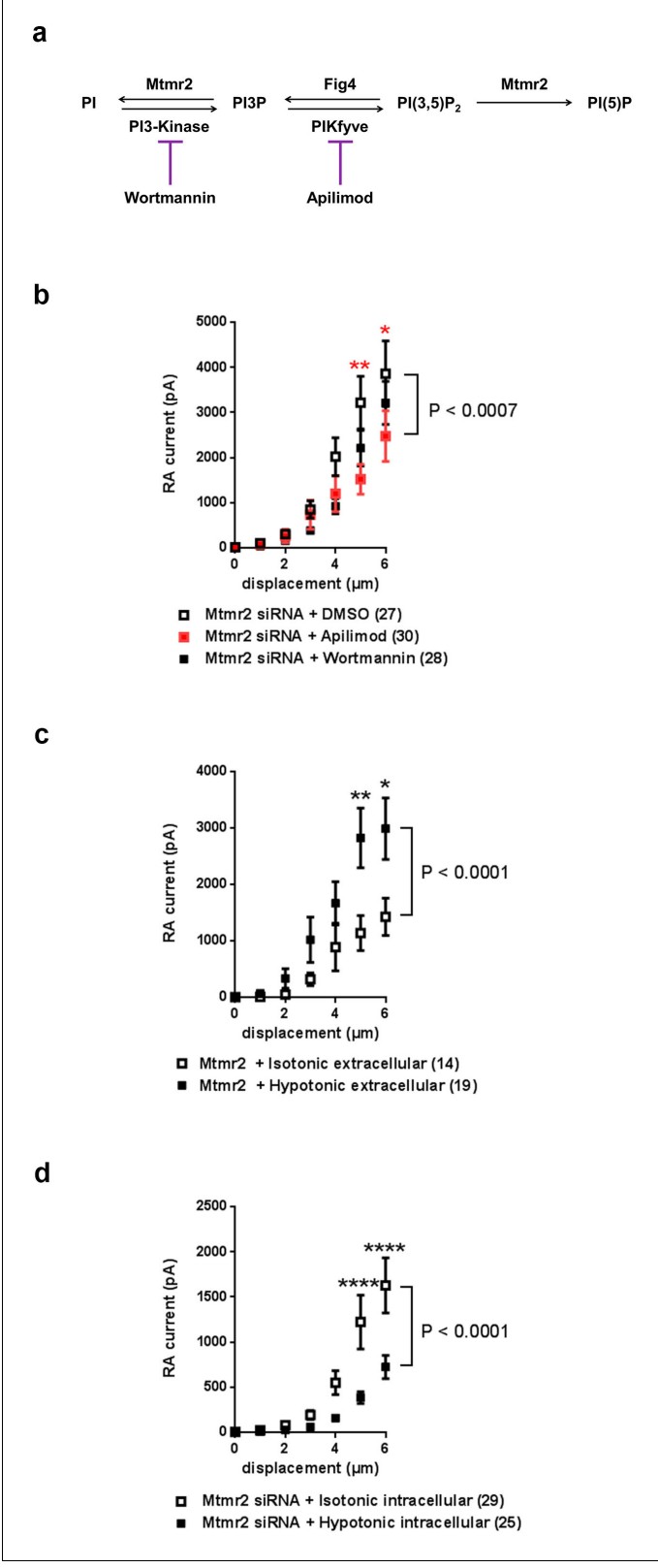

**Figure 4.** Mtmr2 modulates Piezo2-mediated RA-MA currents mainly via PI(3,5)P$_2$. (**a**) Scheme illustrating the major steps of PI(3,5)P$_2$ synthesis and turnover including commonly used inhibitors and their targets. Wortmannin is an inhibitor of the phosphatidylinositol 3-kinase (PI3-Kinase) while Apilimod inhibits phosphatidylinositol 3-phosphate 5-kinase (PIKfyve). The *Fig4* gene encodes a polyphosphoinositide phosphatase. (**b**) Stimulus-current curves after

*Figure 4 continued on next page*

*Figure 4 continued*

addition of Wortmannin, Apilimod or vehicle (DMSO) to *Mtmr2* siRNA-treated neurons (*Mtmr2* siRNA + DMSO: n = 27 neurons; *Mtmr2* siRNA + Wortmannin: n = 28 neurons; *Mtmr2* siRNA + Apilimod: n = 30 neurons). 2-way ANOVA suggested a significant (P<0.0007) overall effect on RA-MA currents. Holm-Sidak's multiple comparisons test was performed to compare both conditions to DMSO at individual stimulus magnitudes. While no significant difference between Wortmannin and DMSO at individual stimulus magnitudes was observed, Apilimod application showed a significant reduction of currents compared to DMSO, p-values are indicated by * in the graph. Similarly, only Apilimod treatment increased the displacement threshold (p=0.0055 compared to DMSO-treated neurons, Kruskal-Wallis test followed by Dunn´s multiple comparisons test; *Supplementary file 1*). The inactivation time constants were unaltered by either treatment (*Supplementary file 1*). (c) Hypotonic extracellular solution counteracted the inhibition of Piezo2 RA-MA currents caused by Mtmr2 overexpression. Stimulus-current curves for Piezo2 RA-MA currents upon extracellular hypotonic stress application to DRG neurons overexpressing Mtmr2 (Mtmr2 + Isotonic extracellular solution: n = 14 neurons; Mtmr2 + Hypotonic extracellular solution: n = 19 neurons; 2-way ANOVA suggested that extracellular hypotonic stress had a significant (P<0.0001) effect on RA-MA currents. Holm-Sidak's multiple comparisons test was performed to compare both conditions at individual stimulus magnitudes, p-values are indicated by * in the graph. The displacement threshold of RA-MA currents and inactivation time constant of RA-MA currents were unchanged (*Supplementary file 1*). (d) Hypotonic intracellular solution counteracted the potentiation of Piezo2 RA-MA currents caused by Mtmr2 knockdown. Stimulus-current curves for Piezo2 RA-MA currents upon intracellular hypotonic stress application to DRG neurons treated with *Mtmr2* siRNA (*Mtmr2* siRNA + Isotonic intracellular solution: n = 29 neurons; *Mtmr2* siRNA + Hypotonic intracellular solution: n = 25 neurons; 2-way ANOVA suggested that intracellular hypotonic stress had a significant (P<0.0001) effect on RA-MA currents. Holm-Sidak's multiple comparisons test was performed to compare both conditions at individual stimulus magnitudes, p-values are indicated by * in the graph. The displacement threshold of RA-MA currents was increased upon intracellular hypotonic stress (p=0.0131; Mann-Whitney test; *Supplementary file 1*). The inactivation time constant of RA-MA currents was unchanged (*Supplementary file 1*).
DOI: https://doi.org/10.7554/eLife.32346.009

The following figure supplements are available for figure 4:

**Figure supplement 1.** Effect on Piezo2 RA-MA currents upon application of PIPs or Apilimod in cultured DRG.
DOI: https://doi.org/10.7554/eLife.32346.010

**Figure supplement 2.** Mtmr2 knockdown does not obviously alter mechanical properties of cultured DRG neurons.
DOI: https://doi.org/10.7554/eLife.32346.011

1). As specific probes to assess the cellular distribution of PI(3,5)P$_2$ are not available (*Hammond et al., 2015*; *Li et al., 2013b*; *McCartney et al., 2014a*; *Nicot and Laporte, 2008*; *Previtali et al., 2007*), we could not measure the actual levels of PI(3,5)P$_2$ in neurons under different osmotic conditions. Even though, our pharmacological and osmotic experiments both suggest an interdependent contribution of Mtmr2 and PI(3,5)P$_2$ to the modulation of Piezo2-mediated MA currents in DRG cultures.

PIPs are essential membrane components and alterations of PIP availability could modify mechanical properties of cells and their membranes (*Janmey, 1995*; *Raucher et al., 2000*; *Skwarekmaruszewska et al., 2006*). In fact, several recent studies demonstrated the influence of membrane mechanics on mechanotransduction (*Brohawn et al., 2014a*; *Brohawn et al., 2014b*; *Cox et al., 2016*; *Lewis and Grandl, 2015*; *Sukharev, 2002*; *Wu et al., 2016*), particularly on Piezo2-mediated RA-MA currents (*Jia et al., 2016*; *Qi et al., 2015*). Therefore, we set out to test a possible impact of Mtmr2 knockdown upon mechanical properties of cultured DRG neurons by atomic force microscopy (AFM) (*Nawaz et al., 2015*; *Qi et al., 2015*; *Rehfeldt et al., 2007*). However, our experiments did not show any significant differences between *Mtmr2* siRNA-treated and control-treated DRG neurons. Neither the Young´s elastic modulus (an indicator for cellular elasticity including membrane tension and cortex stiffness) (*Morley et al., 2016*; *Qi et al., 2015*) as determined from the indentation, nor the tether force (an indicator for membrane tension and mechanical coupling to the cortex) (*Qi et al., 2015*) obtained from the retraction portion of the force distance curves were altered (*Figure 4—figure supplement 2*). These results are in line with our aforementioned findings indicating that Mtmr2 knockdown does not fundamentally alter mechanotransduction in DRG cultures (*Figure 2c,d* and *Supplementary file 1*). Nevertheless, our AFM measurements cannot exclude possible small and local changes in membrane tension in the direct vicinity of Piezo2.

## Piezo2 harbors a PIP$_2$ binding motif

Based on the functional role of PI(3,5)P$_2$ for Mtmr2-dependent Piezo2 regulation we investigated whether Piezo2 can bind PI(3,5)P$_2$. PIPs are known to bind to proteins through various domains such as the FYVE domain, WD40 domain, PHD domain or electrostatically via poly-basic regions with unstructured clusters of positively charged amino acid residues (Lysine or Arginine) (*Dong et al., 2010*; *McCartney et al., 2014a*). Sequence analysis revealed a region in Piezo2 with considerable similarity to the proposed PI(3,5)P$_2$ binding motif of the mucolipin TRP channel 1 (TRPML1) (*Dong et al., 2010*) (*Figure 5a*). We performed a peptide-lipid binding assay to test if this sequence in Piezo2 could bind to PI(3,5)P$_2$, which was indeed the case (*Figure 5b,d*). In addition, we also observed that the Piezo2 peptide was able to bind PI(4,5)P$_2$ and weakly to PI(3,4)P$_2$ (*Figure 5b,d* and *Figure 5—figure supplement 1*). This is an intriguing result because Borbiro and colleagues showed that TRPV1 modulates Piezo2 currents through PI(4,5)P$_2$ depletion, though the study did not report a PI(4,5)P$_2$ binding region in the Piezo2 sequence (*Borbiro et al., 2015*). In parallel we performed the binding assay with a mutated version of the Piezo2 peptide, in which positively-charged amino acid residues shown to be relevant for PI(3,5)P$_2$ binding in TRPML1 were changed to neutral Glutamine (Q; Piezo2 3Q mutant; QQILQYFWMS). This mutated peptide did not bind to any lipid (*Figure 5b,d*). The here identified PI(3,5)P$_2$-binding region in Piezo2 exhibits 50% sequence identity to Piezo1 with conservation of positively charged amino acid residues (*Figure 5a*). Therefore, we also tested whether Piezo1 was able to bind PI(3,5)P$_2$, but did not find any evidence for this (*Figure 5c,d*). These data suggest that not only positively charged amino acid residues, but also flanking amino acids in this Piezo2 region contribute to PIP$_2$ binding in a yet to be explored manner. Further, these in vitro binding studies substantiate our functional data on the specific link between Piezo2 and the Mtmr2 substrate PI(3,5)P$_2$ by identifying a PI(3,5)P$_2$ binding domain in Piezo2, but not in Piezo1. It is important to note that the peptide region defined here may not be the only PI(3,5)P$_2$ binding region in Piezo2 especially when considering the known diversity of PIP modules (*Dong et al., 2010*; *McCartney et al., 2014a*). Due to the sheer size of Piezo2 a large-scale peptide-lipid binding screen was beyond the scope of this study.

Next, we attempted to monitor the functional relevance of the here described PI(3,5)P$_2$ binding domain for the Piezo2-Mtmr2 interaction. To this end we generated a Piezo2 P1 mutant by swapping the PI(3,5)P$_2$ binding domain of Piezo2 with the corresponding domain of Piezo1 (please see scheme in *Figure 5a*). Remarkably, MA currents of the Piezo2 P1 mutant were only slightly attenuated upon co-expression with Mtmr2 compared to mock-transfected controls (*Figure 5e*). Also, the displacement threshold and inactivation time constant of MA currents remained unchanged (*Supplementary file 1*). This is in stark contrast to the pronounced Mtmr2-induced suppression of MA currents recorded from wildtype Piezo2 (*Figure 1f*). Hence, our results suggest that the PI(3,5)P$_2$ binding domain of Piezo2 identified here likely mediates the functional sensitivity of Piezo2 to Mtmr2-dependent changes in PI(3,5)P$_2$ levels.

## Discussion

Vertebrate somatosensory mechanotransduction entails a complex interplay of cellular components; however, their identity is far from being resolved. In our study, we demonstrate that Mtmr2 limits Piezo2 MA currents, and that this effect likely involves local catalysis of PI(3,5)P$_2$. Thus, our work elucidated a link between Mtmr2 and PI(3,5)P$_2$ availability as a previously unappreciated mechanism how Piezo2-mediated mechanotransduction can be locally controlled in peripheral sensory neurons.

Originally, we identified Mtmr2 as a significantly enriched member of the native Piezo2 interactome in DRG neurons (*Narayanan et al., 2016*). Mtmr2 is an active phosphatase that catalyzes the removal of 3-phosphate from its substrates PI(3)P and PI(3,5)P$_2$. Hence, its subcellular localization is crucial as it dictates access to its substrates, which are embedded in membrane lipid bilayers. Interestingly, several reports suggest that membrane association of Mtmr2 is enhanced by hypotonic stress (*Berger et al., 2003*) and also by interactions with other members of the Mtmr family (*Kim et al., 2003*; *Ng et al., 2013*). For example, Mtmr13 and Mtmr2 reciprocally control their abundance at the membrane of cultured cell lines (*Ng et al., 2013*; *Robinson and Dixon, 2005*) and in uncharacterized endomembrane compartments in Schwann cells (*Ng et al., 2013*). In addition, Mtmr2 enzymatic activity is augmented through interaction with Mtmr13 (*Berger et al., 2006*). In this respect it is noteworthy that our interactomics screen (*Narayanan et al., 2016*) identified two

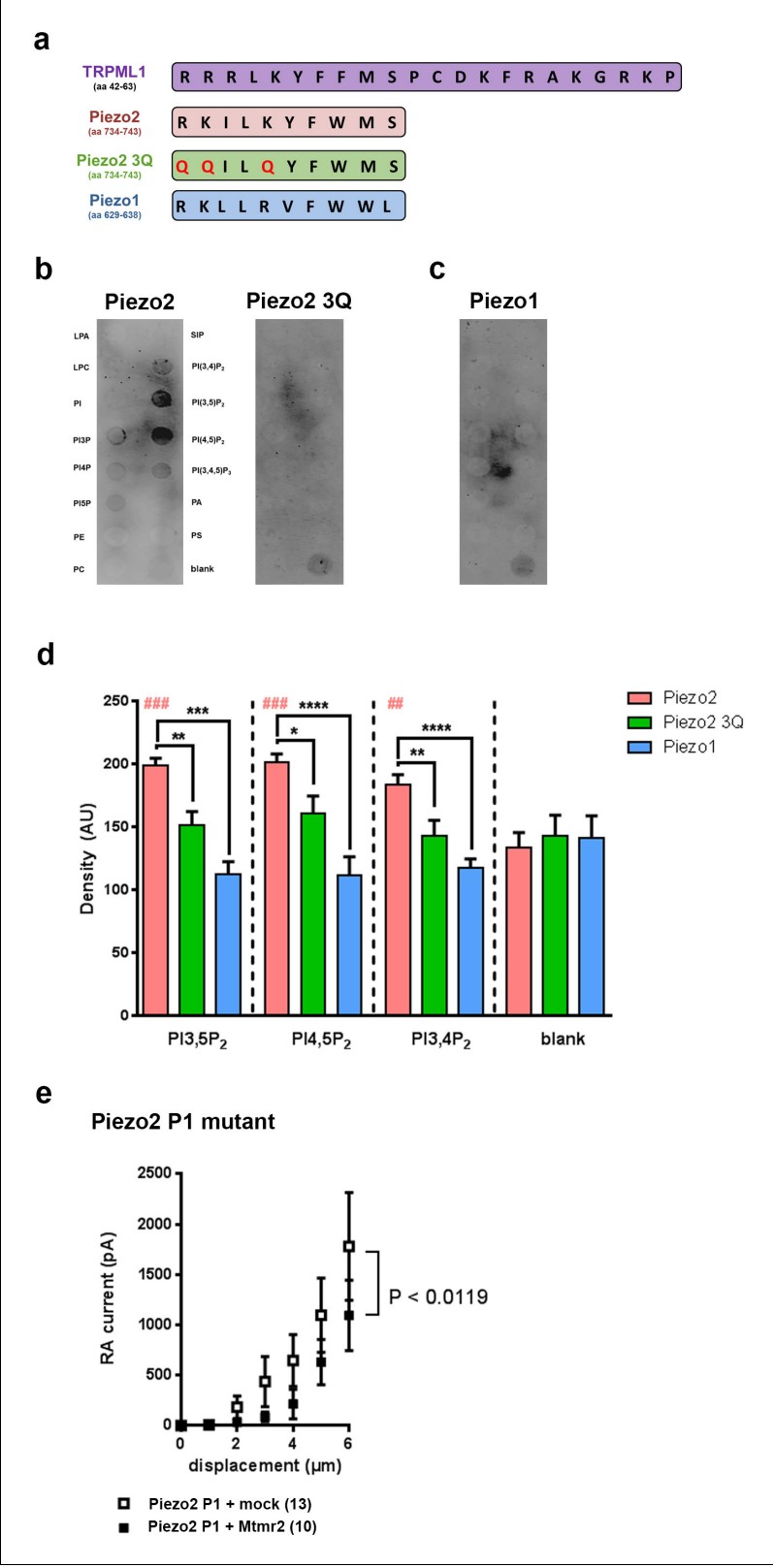

**Figure 5.** Murine Piezo2, but not Piezo1, harbors a PIP$_2$ binding motif. (a) Schematic view of the PI(3,5)P$_2$ binding region of TRPML1 identified elsewhere (**Dong et al., 2010**) and the region in murine Piezo2 that exhibits pronounced sequence similarity to the PI(3,5)P$_2$ binding region of TRPML1. The indicated sequences were used to generate peptides for Piezo2, the Piezo2 3Q mutant and Piezo1. All peptides were tagged with a FLAG-epitope to

*Figure 5 continued on next page*

*Figure 5 continued*

allow for detection with anti-Flag antibodies on immunoblots. (**b–d**) Representative peptide-lipid binding assays followed by immunoblotting after incubation with indicated peptides (**b, c**) and densitometric quantification (**d**). The arrangement of lipids on the lipid-strip is indicated; blank, no lipid was spotted. The Piezo2 peptide strongly binds to PI(3,5)P$_2$ and PI(4,5)P$_2$, and weakly to PI(3,4)P$_2$. Neither the Piezo2 3Q mutant peptide nor the Piezo1 peptide exhibited significant binding to any of the lipids tested. One-way ANOVA followed by Dunnett's multiple comparisons test was used to compare spot signal densities for each peptide to the respective blank, p-values are indicated by # in the graph (**d**). In addition, one-way ANOVA followed by Holm-Sidak's multiple comparisons test was used to compare spot signal densities across the three peptides, p-values are indicated by * in the graph (**d**). The graph only presents data for those lipids, to which the Piezo2 peptide exhibited significant binding. Please see *Figure 5—figure supplement 1* for summarized data on lipids tested with the Piezo2 peptide. Experiments using the Piezo2 peptides were independently repeated 6 times, of which four times were conducted in parallel with experiments using the Piezo1 peptide. AU, arbitrary units. (**e**) Stimulus-current curves upon co-expression of the Piezo2 P1 mutant with Mtmr2 in HEK293 cells compared to mock controls. Mtmr2 only slightly attenuated MA currents (Piezo2 P1 mutant + mock: n = 13 cells; Piezo2 P1 mutant + Mtmr2: n = 10 cells). 2-way ANOVA reported a significant (P<0.0119) overall effect of Mtmr2 overexpression on RA-MA currents of the Piezo2 P1 mutant, however a Holm-Sidak's multiple comparisons test showed no significant difference between currents at individual stimulus magnitudes. Of note, the displacement threshold and inactivation time constant were also unaffected upon overexpression of Mtmr2 compared to mock (*Supplementary file 1*).
DOI: https://doi.org/10.7554/eLife.32346.012

The following figure supplement is available for figure 5:

**Figure supplement 1.** Quantification of peptide-lipid binding assays using the Piezo2 peptide.
DOI: https://doi.org/10.7554/eLife.32346.013

---

additional Mtmr family members: Mtmr1 and Mtmr5. Mtmr1 was shown to be similar to Mtmr2 in structure and substrate specificity (*Bong et al., 2016*), but is much less studied than Mtmr2. Mtmr5 is a catalytically inactive phosphatase which binds to Mtmr2, increases its enzymatic activity and controls its subcellular localization (*Kim et al., 2003*). The fact that Mtmr5 was previously reported as a binding partner of Mtmr2 further validates our published interactomics data (*Narayanan et al., 2016*), and hints towards the intriguing possibility that a multiprotein complex of different Mtmr family members might affect Piezo2 function. Future experiments should focus on assessing the role of these two Mtmr family members for mechanotransduction in peripheral sensory neurons.

Mechanistically, our in vitro data strongly suggest that enzymatic activity of Mtmr2 and consequently PI(3,5)P$_2$ availability modulates Piezo2 RA-MA currents. Several lines of evidence support this conclusion. We found that, unlike wild type Mtmr2, a catalytically inactive mutant of Mtmr2 (C417S) did not suppress Piezo2 currents in HEK293 cells. In sensory neurons expressing *Mtmr2* siRNA, we could counteract RA-MA current potentiation by inhibiting PI(3,5)P$_2$ synthesis in two ways: via Apilimod (*Laporte et al., 1998*; *McCartney et al., 2014b*; *Mironova et al., 2016*; *Previtali et al., 2007*; *Vaccari et al., 2011*) and via intracellular hypoosmolarity (*Dove et al., 1997*; *McCartney et al., 2014a*), respectively. In analogy, increasing PI(3,5)P$_2$ levels by extracellular hypoosmolarity (*Berger et al., 2003*; *Dove et al., 1997*) attenuated the suppression of RA-MA currents upon Mtmr2 overexpression. Here, it is important to note that Mtmr2 activity usually serves a dual function: dephosphorylation of PI(3,5)P$_2$ and of PI(3)P, albeit the latter with lower efficiency (*Berger et al., 2002*). Yet, our results upon application of Wortmannin suggest only a marginal contribution of PI(3)P to altering Piezo2-mediated mechanotransduction. Wortmannin is commonly used to block class III PI 3-kinase (please see scheme in *Figure 4a*), however, it also is reported to inhibit class I PI 3-kinase (*Messenger et al., 2015*). Hence its action on class III PI 3-kinase might not have been efficient enough in our experiments. Biologically, many enzymes control PI(3)P synthesis (*Yan and Backer, 2007*; *Zolov et al., 2012*) and previous work on Mtmr2 has only shown minor modifications of PI(3)P levels in Mtmr2-deficient cells (*Cao et al., 2008*; *Vaccari et al., 2011*) in line with its substrate preference (*Berger et al., 2002*). Moreover, our lipid-peptide binding assays demonstrated that, in a cell-free system, a distinct region in Piezo2, which is similar to the known PI(3,5)P$_2$-binding motif of TRPML1, can bind PI(3,5)P$_2$. Therefore, our data are highly indicative of a prominent involvement of the Mtmr2 substrate PI(3,5)P$_2$ in controlling somatosensory mechanosensitivity. Whether or not the expected change of PI(5)P after PI(3,5)P$_2$ catalysis (please see scheme in

*Figure 4a*) plays a role could not be investigated due to the lack of tools and knowledge about PI(5) P-specific physiology (*McCartney et al., 2014a*; *Zolov et al., 2012*).

Interestingly, the here identified PI(3,5)$P_2$-binding region is conserved among Piezo2 in vertebrates and exhibits 50% of sequence identity to mouse Piezo1 with conservation of basic amino acid residues. Arthropods and nematodes, which only encode one Piezo protein, exhibit lower sequence similarity in this region, 40% (D. melanogaster) and 30% (C. elegans), respectively. It would be of high interest to determine in future studies whether other members of the Piezo family also bind PI (3,5)$P_2$ and are potentially regulated by Mtmr2 or its homologs. Yet, in mice the here described PI (3,5)$P_2$-binding and inhibition via Mtmr2 seemed quite specific for Piezo2 as indicated by our peptide-lipid binding assays and functional experiments on the domain-swapped Piezo2 P1 mutant. In contrast to the majority of studies on the regulation of vertebrate mechanotransduction (*Borbiro et al., 2015*; *Eijkelkamp et al., 2013*; *Morley et al., 2016*; *Poole et al., 2014*; *Qi et al., 2015*), we did not observe alterations in Piezo1-mediated MA currents in HEK293 cells, and the magnitude of IA- and SA-MA currents was largely unaffected in DRG cultures. Surprisingly though, we detected a mild increase in the proportion of RA-MA cells paralleled by a decrease of the IA-MA subpopulation. While there is some evidence that Piezo2 might also play a role for IA-MA currents (*Lou et al., 2013*), to date the molecular identity of IA-MA currents has not been resolved (*Viatchenko-Karpinski and Gu, 2016*). For that reason the interpretation of this finding awaits further clarification of the molecular nature of IA-MA currents.

In principle, the regulation of Piezo2 by lipids is not unexpected and our data significantly advance our knowledge about the link between mechanotransduction and components of the lipid bilayer (*Brohawn et al., 2014a*, *2014b*; *Cox et al., 2016*; *Lewis and Grandl, 2015*; *Sukharev, 2002*; *Wu et al., 2016*). In particular, PI(4,5)$P_2$ and its precursor PI(4)P (*Borbiro et al., 2015*) as well as cholesterol (*Qi et al., 2015*) were recently found to be implicated in mechanotransduction mediated by both, Piezo1 and Piezo2. In contrast to PI(4,5)$P_2$, our understanding of PI(3,5)$P_2$ function, localization and regulation is limited to date (*McCartney et al., 2014a*). PI(3,5)$P_2$ is much less abundant than most PIPs, for example 125-fold less than PI(4,5)$P_2$ in mammalian cells (*Zolov et al., 2012*), and tightly regulated by a large protein complex (*Vaccari et al., 2011*; *Zolov et al., 2012*) (please see scheme in *Figure 4a*). PI(3,5)$P_2$ was believed to predominantly act in the endo-lysosome system of eukaryotes (*Di Paolo and De Camilli, 2006*; *Dong et al., 2010*; *Ho et al., 2012*; *Zolov et al., 2012*). By now the picture is emerging that PI(3,5)$P_2$ can serve a diverse range of cellular functions, such as autophagy, signaling in response to stress, control of membrane traffic to the plasma membrane as well as regulation of receptors and ion channels (*Dong et al., 2010*; *Ho et al., 2012*; *Ikonomov et al., 2007*; *Klaus et al., 2009*; *McCartney et al., 2014b*; *Tsuruta et al., 2009*; *Zhang et al., 2012*). Moreover, depending on the cell type, PI(3,5)$P_2$ synthesis and metabolism are dynamically regulated and subject to cellular stimulation and stressors, for example insulin-mediated PI(3,5)$P_2$ changes in adipocytes (*Ikonomov et al., 2007*), neuronal activity-dependent synthesis at hippocampal synapses (*McCartney et al., 2014b*; *Zhang et al., 2012*) and the here exploited regulation of PI(3,5)$P_2$ upon osmotic shock in mammalian cell lines (*Dove et al., 1997*; *Nicot and Laporte, 2008*; *Previtali et al., 2007*).

These reports nourish the notion that changes of PI(3,5)$P_2$ via Mtmr2 might contribute to regulating Piezo2 and, by extension, touch sensitivity. How could this be achieved mechanistically? Instead of acting in a cell-wide manner, PI(3,5)$P_2$ synthesis and turnover is confined to membrane microdomains. Despite their unknown composition and subcellular localization, these microdomains are believed to ensure dynamic and local control of PI(3,5)$P_2$ levels (*Ho et al., 2012*; *Jin et al., 2016*; *McCartney et al., 2014a*). Concomitantly, the abundance of downstream effector proteins is likely to be altered, as well (*Ho et al., 2012*; *Jin et al., 2016*; *McCartney et al., 2014a*). Mtmr2 may physically bind to Piezo2 in order to ensure its enrichment in PI(3,5)$P_2$ microdomains, so that local PI(3,5)$P_2$ changes or yet to be identified effector proteins can efficiently modulate Piezo2 (*Figure 6*, working model). In these microdomains Mtmr2 would catalytically decrease PI(3,5)$P_2$ levels, which in turn could inhibit Piezo2 function. Hence, a physical interaction between Mtmr2 and Piezo2 would selectively concentrate Piezo2 at the sites of PI(3,5)$P_2$ depletion thereby allowing its inhibition in a defined membrane compartment. On the contrary, reduced Mtmr2 expression or activity would be expected to increase local PI(3,5)$P_2$ levels (*Laporte et al., 1998*; *McCartney et al., 2014b*; *Mironova et al., 2016*; *Previtali et al., 2007*; *Vaccari et al., 2011*) and facilitate Piezo2 potentiation. It is conceivable that Mtmr2-controlled PI(3,5)$P_2$ availability could in turn cause local changes in

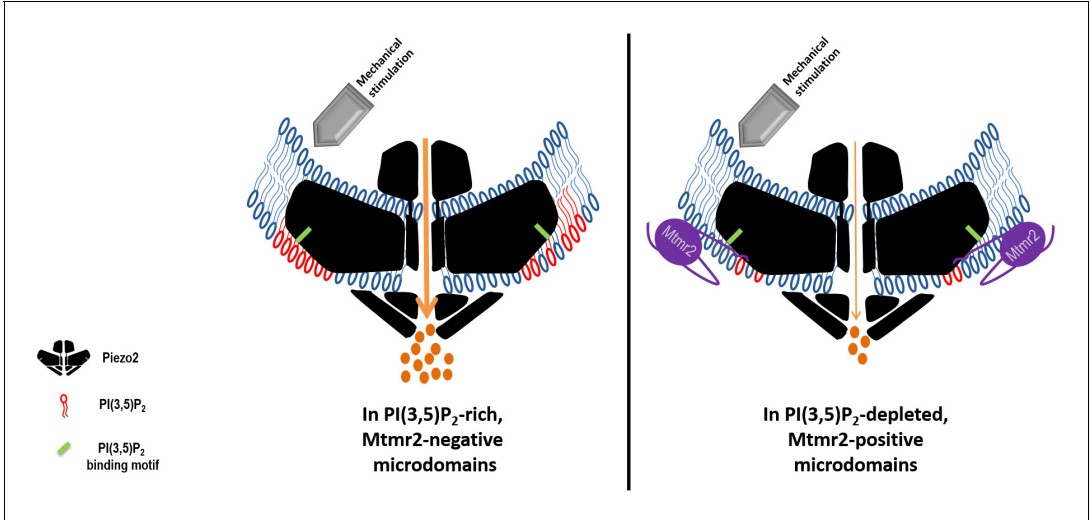

**Figure 6.** Working model: Local control of Piezo2 function by interdependent actions of Mtmr2 and PI(3,5)P$_2$. Mtmr2 controls the abundance of PI(3,5)P$_2$ by dephosphorylation (please see *Figure 4a*). Mtmr2 and Piezo2 expression as well as PI(3,5)P$_2$ might be compartmentalized in membrane microdomains. Piezo2 localization in Mtmr2-negative microdomains would facilitate its access to local PI(3,5)P$_2$ and consequently potentiate Piezo2 RA-MA currents (left side). On the other hand, high Mtmr2 levels and its localization in the proximity of Piezo2 would augment PI(3,5)P$_2$ turnover, thereby decreasing local PI(3,5)P$_2$ availability and suppressing Piezo2 RA-MA currents (right side). One could further speculate that Mtmr2, via binding Piezo2, might recruit Piezo2 to membrane microdomains depleted of PI(3,5)P$_2$. This would provide an active mechanism to inhibit Piezo2 RA-MA currents in membrane compartments – may they be at the plasma membrane or intracellular membranes. Ultimately, Mtmr2 and PI(3,5)P$_2$ may contribute to dynamically tuning touch sensitivity of an organism in response to diverse conditions modulating Mtmr2 and PI(3,5)P$_2$ levels (e.g. osmotic stress as indicated by results shown in *Figure 4*).The following questions await further clarification: (i) How are Mtmr2/PI(3,5)P$_2$ regulated during (patho) physiological conditions in the somatosensory system, (ii) does the modulation of RA-MA currents require additional yet to be identified effector proteins, and (iii) are other PIPs also involved, for example PI(4,5)P$_2$ shown to modulate Piezo2 function (*Borbiro et al., 2015*)? As the structure of Piezo2 has not been resolved yet, Piezo2 is depicted after the recently solved structure of Piezo1 (*Ge et al., 2015*; *Guo and MacKinnon, 2017*; *Saotome et al., 2018*). If this structure holds also true for Piezo2, the PI(3,5)P$_2$ binding domain (depicted in green) would roughly be localized within the first third of the N-terminal blade.

DOI: https://doi.org/10.7554/eLife.32346.014

membrane tension (*Lewis and Grandl, 2015*; *Perozo et al., 2002*; *Vásquez et al., 2014*; *Wu et al., 2017*). The latter has already been demonstrated to affect Piezo1 MA currents (*Cox et al., 2016*; *Lewis and Grandl, 2015*; *Wu et al., 2016*). Whether Piezo2 is also sensitive to local membrane tension and whether PI(3,5)P$_2$ levels indeed influence membrane tension in sensory neurons remains to be investigated. Unfortunately, local changes in membrane tension in the vicinity of Piezo2 are likely too small to be captured by our AFM-based measurements.

Nevertheless, the data presented here allow us to infer an attractive mechanism exquisitely suited for transient and compartmentalized control of Piezo2 function, that is physical vicinity to Mtmr2, the activity status of Mtmr2 and consequently PI(3,5)P$_2$ availability (*Figure 6*). We can further speculate that this local control of Piezo2 function may serve to tune the threshold and magnitude of neuronal activity to light touch. Ultimately, Mtmr2 and PI(3,5)P$_2$ may represent a means by which touch sensitivity of an organism can be dynamically adjusted in response to diverse stimuli modulating Mtmr2 and PI(3,5)P$_2$ levels (*Figure 6*). In this respect it is noteworthy that multiple Mtmr2 mutations – including those affecting its activity (*Berger et al., 2002*) – have been implicated in Charcot-Marie-Tooth type 4B1 (CMT4B1) disease, a peripheral neuropathy characterized by abnormalities in myelination and nerve conduction (*Bolino et al., 2004*; *Bolis et al., 2005*; *Bonneick et al., 2005*). An analysis of Piezo2 mechanotransduction, light touch as well as tactile hypersensitivity in Mtmr2 knockout (*Bolino et al., 2004*) and Mtmr2 mutant mice (*Bonneick et al., 2005*) would be warranted to thoroughly examine the potential role of Mtmr2 for (patho)physiological aspects of vertebrate mechanosensation.

Several important questions remain to be investigated. While we identified a PI(3,5)P$_2$ binding motif in Piezo2 in vitro, we can neither gauge the affinity of the fully assembled Piezo2 trimer for PI(3,5)P$_2$ nor assess whether PI(3,5)P$_2$ and Mtmr2 bind allosterically, competitively or independently of

each other. Moreover, our peptide-lipid binding assay shows that the extremely abundant $PI(4,5)P_2$ and to a lesser extent $PI(3,4)P_2$ could bind the same motif as $PI(3,5)P_2$. This raises the question how $PIP_2$ specificity can be achieved by Piezo2 when embedded in cellular membranes. Piezo2 might be differentially localized in membranes dependent on $PI(3,5)P_2$ levels and Mtmr2 abundance. We did not observe any differences in the overall abundance of Piezo2 channels at the plasma membrane of sensory neurons. Yet, given that $PI(3,5)P_2$ acts in membrane microdomains, it would be desirable to visualize whether Piezo2 is localized in subcellular membrane compartments, that is (i) in $PI(3,5)P_2$-enriched versus $PI(3,5)P_2$-depleted and/or (ii) Mtmr2-harboring versus Mtmr2-negative microdomains. It remains to be seen whether these membrane compartments are confined to the plasma membrane and/or to intracellular membranes such as endo-lysosomes, which are known to contain the majority of $PI(3,5)P_2$ (*Di Paolo and De Camilli, 2006*; *Dong et al., 2010*; *Zolov et al., 2012*). Along these lines it is worth mentioning that AMPA receptor abundance at hippocampal synapses has been shown to be regulated by $PI(3,5)P_2$-controlled cycling through early and late endosomes (*McCartney et al., 2014b*). While unknown so far, localization of Piezo2 to intracellular membranes would not be unexpected (*Coste et al., 2010*) since its family member Piezo1 has originally been described to reside in the endoplasmic reticulum (*McHugh et al., 2010*). Thus, exploring endocytosis and intracellular trafficking of Piezo2 may offer novel insights into its regulation by the intracellular membrane pool of PI(3,5)P2.

To address some of these issues the development of high-affinity Mtmr2 and Piezo2 antibodies as well as $PI(3,5)P_2$-specific probes (*Hammond et al., 2015*; *Li et al., 2013b*; *McCartney et al., 2014a*; *Nicot and Laporte, 2008*; *Previtali et al., 2007*), which faithfully represent their subcellular spatial and temporal dynamics, would be required. In contrast to $PI(4,5)P_2$, such probes have not yet been successfully designed for $PI(3,5)P_2$, and the value of the few existing probes is questionable due to spatial restrictions and limited specificity for $PI(3,5)P_2$ (*Hammond et al., 2015*; *Li et al., 2013b*; *McCartney et al., 2014a*; *Nicot and Laporte, 2008*; *Previtali et al., 2007*). Future studies using these tools combined with high-resolution microscopy have the potential to address fundamental aspects of Piezo2 trafficking, localization, and function.

Taken together, our data present Mtmr2 as a novel modulator of the mechanosensory apparatus, and we provide evidence for the functional convergence of Mtmr2 enzymatic activity and $PI(3,5)P_2$ availability onto Piezo2-mediated mechanotransduction in peripheral sensory neurons. In essence, we propose the Mtmr2-Piezo2 interaction as a previously unappreciated mechanism to locally and dynamically regulate Piezo2 function and, consequently, the organism´s response to light tough. Therefore, our study significantly advances our understanding of the complex molecular machinery underlying somatosensory mechanosensitivity in vertebrates.

## Materials and methods

### Key resources table

| Reagent type (species) or resource | Designation | Source or reference | Identifiers | Additional information |
|---|---|---|---|---|
| Strain, strain background (mouse) | B6/J mice | | RRID: IMSR_JAX:000664 | bred in the animal facility of the MPIem Goettingen |
| Strain (mouse) | Piezo2GFP | kind gift of Ardem Patapoutian | | bred in the animal facility of the MPIem Goettingen |
| Cell line (human) | HEK293 | purchased from ATCC | RRID: CVCL_0045 | Cells were not tested for mycoplasma contamination; cells were authenticated by ATCC upon purchase |
| Antibody | Rabbit anti-Mtmr2 (1:100) | Biotechne, #NBP1-33724 | RRID: AB_2147841 | |
| | Chicken anti-GFP (1:500) | Thermo Fisher Scientific, #A10262 | RRID: AB_2534023 | |
| | Rabbit anti-GST (1:500) | Santa Cruz, #sc-459 | | |

*Continued on next page*

*Continued*

| Reagent type (species) or resource | Designation | Source or reference | Identifiers | Additional information |
|---|---|---|---|---|
| | Mouse anti-myc (1:750, 1:500) | Santa Cruz, #sc-47694 | RRID: AB_627266 | |
| | Mouse anti-FLAG (1:100) | Sigma Aldrich, #F1804 | RRID: AB_262044 | |
| | Rabbit anti-Piezo2 (1:200) | Novus Biologicals, #NBP1-78624 | RRID: AB_11005294 | |
| Recombinant DNA reagent | pCMVSport6 Piezo2-GST IRES GFP | kind gift of Ardem Patapoutian | mouse *Piezo2* | |
| | pCMV6-Entry Mtmr2-myc-DDK | Origene, #MR215223 | mouse *Mtmr2* | |
| | Mtmr2C417S-myc-DDK | | mouse *Mtmr2 C417S* | Mutation generated using Q5 Site-Directed Mutagenesis kit (New England BioLabs) |
| | pCMV Sport6 Piezo1-753-myc-IRES GFP | kind gift of Ardem Patapoutian | mouse *Piezo1* | Myc tag was inserted at amino acid 753 as described in ***Coste et al., 2015***. |
| | pGEM-Teasy Kv1.1-HA | | mouse *Kv1.1* | Custom-made and sequence-verified |
| | pCMVSport6 | kind gift of Ardem Patapoutian | | |
| | pCDNA3.1-myc-His | Invitrogen, #V80020 | | |
| | pCNDA3-GST | kind gift of Ardem Patapoutian | | |
| | pCMVSport6 Piezo2 P1 mutant-GST IRES GFP | | mouse *Piezo2 P1* mutant | Mutation generated using Q5 Site-Directed Mutagenesis kit (New England BioLabs) |
| | pCDNA3.1-myc-His TRPA1 | kind gift of Ardem Patapoutian | mouse TRPA1 | |
| | pCMV6-Vti1b-myc-DDK | Origene | | |
| Sequence-based reagent | *Mtmr2* forward primer for qPCR | MPIem DNA Core Facility | TGTACCCCACCATTGAAGAAA | |
| | *Mtmr2* reverse primer for qPCR | MPIem DNA Core Facility | TAAGAGCCCCTGCAAGAATG | |
| | *Piezo2* forward primer for qPCR | MPIem DNA Core Facility | AGGCAGCACATAGGATGGAT | |
| | *Piezo2* reverse primer for qPCR | MPIem DNA Core Facility | GCAGGGTCGCTTCAGTGTA | |
| | *Actb* forward primer for qPCR | MPIem DNA Core Facility | GATCAAGATCATTGCTCCTCCTG | |
| | *Actb* reverse primer for qPCR | MPIem DNA Core Facility | CAGCTCAGTAACAGTCCGCC | |
| | *Gapdh* forward primer for qPCR | MPIem DNA Core Facility | CAATGAATACGGCTACAGCAAC | |
| | *Gapdh* reverse primer for qPCR | MPIem DNA Core Facility | TTACTCCTTGGAGGCCATGT | |
| | *Piezo2* mutagenesis forward primer | MPIem DNA Core Facility | GTCTTCTGGTGGCTCGTGGT CATTTATACCATGTTGG | |
| | *Piezo2* mutagenesis reverse primer | MPIem DNA Core Facility | ACGCAGCAGCTTCCTCCACC ACTCGTAGTGCAC | |
| | *Mtmr2* mutagenesis forward primer | MPIem DNA Core Facility | GTGGTACACTCCAGTGATGGATG | |
| | *Mtmr2*mutagenesis reverse primer | MPIem DNA Core Facility | CACAGACGTCTTCCCAGA | |

*Continued on next page*

*Continued*

| Reagent type (species) or resource | Designation | Source or reference | Identifiers | Additional information |
|---|---|---|---|---|
| Peptide, recombinant protein | Piezo2-FLAG tagged | Custom-made by GenScript | EWWRKILKYFWMSVVIDYKD DDDKQNN | |
| | Piezo2 3Q-FLAG tagged | Custom-made by GenScript | EWWQQILQYFWMSVVIDYKD DDDKQNN | |
| | Piezo1-FLAG tagged | Custom-made by GenScript | TLWRKLLRVFWWLVDYKDDD DKQNN | |
| Chemical compound, drug | Wortmannin | Sigma Aldrich | | |
| | Apilimod | Bertin Pharma | | |
| | PI(3,5)P2 | Echelon | | |
| | PI(3)P | Echelon | | |
| Software, algorithm | Fitmaster | HEKA Electronik GmbH | | |
| | Patchmaster | HEKA Electronik GmbH | | |
| | ImageJ | NIH (*Schindelin et al., 2015*) | RRID: SCR_003070 | |
| | GraphPad Prism 6.01 | GraphPad Software | RRID: SCR_015807 | |

## DRG culture and transfection

Preparation and culture of mouse DRG neurons were performed as described previously (*Coste et al., 2010*; *Narayanan et al., 2016*). Throughout the study, DRG were isolated from 9 to 10 week old male C57BL/6J mice or, in case of experiments in *Figure 1a*, *Figure 1—figure supplement 1a,c*, *Figure 2—figure supplement 1c,d* from *Piezo2*[GFP] mice (*Woo et al., 2014*). In brief, DRG neurons were promptly isolated and digested with collagenase (Thermo Fisher Scientific, Germany) and papain (Worthington, Lakewood, USA). Neurons were plated on poly-D-lysine (1 mg/mL, Merck Millipore, Germany) coated coverslips, which were additionally coated with laminin (20 µg/mL, Thermo Fisher Scientific). Growth medium (Hams F12/DMEM 1:1 ratio with L-glutamine; Gibco, Germany) was supplemented with 10% horse serum (Thermo Fisher Scientific) and 100 ng/ml NGF, 50 ng/ml GDNF, 50 ng/ml BDNF, 50 ng/ml NT-3, and 50 ng/ml NT-4 (all growth factors were procured from R&D Systems, Germany).

Transfection of neurons was achieved by nucleofection of siRNA or plasmid into freshly isolated DRG neurons using the P3 Primary Cell 4D Nucleofector X Kit with the 4D-Nucleofector X Unit according to the manufacturer's instructions (Lonza, Germany). 500 nM FlexiTube GeneSolution *Mtmr2* siRNA (Qiagen, #GS77116, Germany) or AllStar Negative control siRNA (CTRL, Qiagen, #SI03650318) for knockdown in DRG neurons and 0.5 µg of pCMV6 *Mtmr2-myc-DDK* (Origene) or 0.5 µg of pCMVSport6 or pcDNA3.1 myc-His (CTRL) for overexpression in DRG neurons, was used. After nucleofection, neurons were allowed to recover in calcium free RPMI medium (Thermo Fisher Scientific) for 10 min at 37°C before plating in growth medium. Two hours after transfection half of the growth medium was exchanged with fresh medium and neurons were grown for 48–72 hr before being used for electrophysiology, immunostaining or qPCR.

## HEK293 cell culture and transfection

HEK293 cells were authenticated by ATCC (Manassas, USA) upon purchase. Thereafter, cell line identity was authenticated by regular morphological inspection. Symptoms for mycoplasma contamination were not observed and thus no test for mycoplasma contamination was performed. Cells were cultured in DMEM with Glutamax (Thermo Fisher Scientific) supplemented with 10% FBS (Fetal bovine serum, Thermo Fisher Scientific) and 5% Pen/Strep (Thermo Fisher Scientific). Cells were grown up to 80–90% confluence before being used for transfection. Transfection was done using Fugene HD Transfection reagent (Promega, Germany). Cells were plated on poly-D-lysine-coated coverslips and maintained in culture for 48 hr before being used for electrophysiology or proximity ligation assay (PLA).

## cDNA and plasmids

pCMVSport6 *Piezo2-GST*-IRES GFP (kind gift from Prof. Ardem Patapoutian, La Jolla, USA); pCMV6-Entry *Mtmr2-myc-DDK* (Origene, #MR215223); pCMV6-Entry *Mtmr2C417S-myc-DDK* (mutation as described in (*Berger et al., 2002*)); pCMV Sport6 *Piezo1-753-myc*-IRES GFP (kind gift from Ardem Patapoutian) (*Coste et al., 2015*); pGEM-Teasy *Kv1.1-HA*; pCMVSport6; pCDNA3.1-myc-His (Invitrogen, #V80020); pCNDA3-GST (kind gift from Prof. Ardem Patapoutian); pCMVSport6 *Piezo2 P1 mutant-GST*-IRES GFP. *Mtmr2C417S* mutant and *Piezo2 P1* mutant were prepared using Q5 Site-Directed Mutagenesis kit (New England BioLabs). Mutagenesis was done according to manufactures instructions and all mutant plasmids were verified by sequencing. Primers used for the mutagenesis were as follows: *Mtmr2* mutagenesis forward primer: GTGGTACACTCCAGTGATGGATG; *Mtmr2* mutagenesis reverse primer: CACAGACGTCTTCCCAGA; *Piezo2* mutagenesis forward primer: GTCTTCTGGTGGCTCGTGGTCATTTATACCATGTTGG; *Piezo2* mutagenesis reverse primer: ACGCAGCAGCTTCCTCCACCACTCGTAGTGCAC.

## Quantitative PCR (qPCR)

Total RNA was isolated from cultured DRG neurons (transfected with *Mtmr2* siRNA or CTRL, please see above), using NucleoSpin RNA XS (Macherey-Nagel) according to the manufacturer's instructions. First-strand cDNA synthesis was done using QuantiTect reverse transcription kit (Qiagen). Mtmr2 and Piezo2 gene expression was assessed in both conditions by real-time qPCR using the SYBR green system (Power SYBR Green PCR Master Mix; Thermo Fisher Scientific) on a LightCycler 480 instrument (Roche, Germany). The melting curve analysis of amplified products was used to confirm the specificity of qPCR assay. All samples were run in triplicate and negative control reactions were run without template. Threshold cycle (Ct) values, the cycle number in which SYBR green fluorescence rises above background, were normalized to two reference genes (*Actb* and *Gapdh*) and recorded as a measure of initial transcript amount. Relative quantification was performed using the 'fit point' as well as the 'second derivative maximum' method of the LightCycler 480. Primer sequences 5'−3' are the following: *Mtmr2* (fw: tgtaccccaccattgaagaaa; rev: taagagccctgcaagaatg), *Piezo2* (fw: aggcagcacataggatggat; rev: gcagggtcgcttcagtgta), *Actb* (fw: gatcaagatcattgctcctcctg; rev: cagctcagtaacagtccgcc), *Gapdh* (fw: caatgaatacggctacagcaac; rev: ttactccttggaggccatgt). Of note, only data normalized to *Actb* are shown but data normalized to *Gapdh* gave similar results. Our qPCR results indicate successful siRNA-mediated knockdown of Mtmr2 across the whole coverslip, which also includes non-transfected neurons and glia cells. Therefore our data do not report on the transfection efficiency and extent of Mtmr2 knockdown in individual neurons.

## Electrophysiology

Whole-cell voltage clamp recordings were performed in transfected DRG cultures, wild type DRG cultures or transfected HEK293 cell cultures at room temperature as described in (*Narayanan et al., 2016*). Briefly, (protocol adapted from [*Coste et al., 2010*]) to elicit mechanically activated currents, the cell soma was mechanically stimulated using a blunt probe (fire polished borosilicate glass capillary). The stimulation was delivered using a piezo-electrically driven micromanipulator (Physik Instrumente GmbH and Co.KG, Germany). The probe was initially positioned ~4 µm from the cell body and had a velocity of 0.8 µm/ms during the ramp phase (forward motion). The stimulus was applied for 150 ms with an inter-stimulus interval of 180 ms. Stimulus-current measurements were performed using mechanical stimulations from 0 to 6 µm in 1 µm increments at a holding potential of −70 mV in whole cell mode. All recordings other than specifically indicated were made in standard extracellular solution containing (in mM) 127 NaCl, 3 KCl, 1 $MgCl_2$, 2.5 $CaCl_2$, 10 Glucose and 10 HEPES; pH = 7.3; osmolarity = 285 mOsm (*Coste et al., 2010*). To achieve hypotonicity the extracellular solution was adjusted to 160 mOsm. The intracellular solution for DRG neurons contained (in mM) 133 CsCl, 10 HEPES, 5 EGTA, 1 $MgCl_2$, 1 $CaCl_2$, 4 MgATP and 0.4 NaGTP; pH = 7.3; osmolarity = 280 mOsm (*Coste et al., 2010*). Hypotonic intracellular solution contained (in mM) 65 CsCl, 10 HEPES, 5 EGTA, 1 $MgCl_2$, 1 $CaCl_2$, 4 MgATP and 0.4 NaGTP; pH = 7.3; osmolarity = 162 mOsm. Intracellular solution for HEK293 cells contained (in mM) 110 KCl, 10 NaCl, 1 $MgCl_2$, 1 EGTA and 10 HEPES; pH = 7.3 (*Poole et al., 2014*). Based on protocols published elsewhere (*Jia et al., 2016*) recordings in extracellular hypotonic solutions were performed as follows: Cells were incubated with the hypotonic solution for 5 min at 37°C prior to recording, after which RA-MA currents were recorded in

hypotonic extracellular solution between 0–20 min after initial application. For some experiments described in this study, DRG neurons were treated with chemical inhibitors of the phosphatidylinositol phosphate pathway. Cells were treated with 35 µM Wortmannin (Sigma Aldrich; protocol adapted from [*Mo et al., 2009*]) in DMSO or 1 µM Apilimod (Bertin Pharma, Germany ; protocol adapted from [*Mironova et al., 2016*]) in DMSO for 2 hr prior to recording. DMSO (0.08% for Apilimod and 0.35% for Wortmannin), was used as vehicle for each experiment. For PIP addition experiments, 1 µM PI(3,5)P$_2$ (Echelon) or 1 µM PI(3)P (Echelon) were included in the isotonic intracellular solution (protocol adapted from [*Dong et al., 2010*]). Of note, the water-soluble diC8 form of the lipids was used as indicated elsewhere (*Dong et al., 2010*).

Data analysis and representation was done using Fitmaster (HEKA Electonik GmbH, Germany) and Igor Pro 6.37 (WaveMetrics, USA). The current magnitude was calculated by measuring peak amplitudes (at each stimulus point) and subtracting leak current.

The displacement threshold was defined as the minimum displacement required to elicit a visible RA-MA current, that is the displacement at which current values exceeded 100 pA in DRG (*Jia et al., 2016*; *Narayanan et al., 2016*) and 50 pA in HEK293 cells (*Szczot et al., 2017*). In HEK293 cells, this cutoff also served to prevent contamination of recordings by HEK293 endogenous Piezo1 (*Dubin et al., 2017*; *Szczot et al., 2017*). Of note, responses in HEK293 cells increased proportionally to stimulus strength and were absent in cells not transfected with *Piezo1* or *Piezo2* plasmids.

For measurements of inactivation kinetics current traces reaching at least 75% of maximal current amplitude were fitted with a mono-exponential or bi-exponential equation and the fast time constant was used for analysis (*Coste et al., 2010*). Data for the displacement threshold and inactivation time constant (τ) are provided in *Supplementary file 1*.

In DRG cultures neuronal soma size differs considerably within a coverslip. Therefore neurons have been traditionally categorized into small, medium and large diameter neurons (*Drew et al., 2004*; *Hu and Lewin, 2006*; *Patapoutian et al., 2009*; *Woo et al., 2014*)exhibiting well-reported differences in function and MA currents (*Drew et al., 2004*; *Hu and Lewin, 2006*; *Patapoutian et al., 2009*; *Poole et al., 2014*; *Woo et al., 2014*). To account for variability of our data due to soma size, we (i) recorded RA currents from visibly large diameter neurons (cell capacitance >40 pF) likely representing mechanoreceptors (*Drew et al., 2004*; *Hu and Lewin, 2006*; *Poole et al., 2014*; *Woo et al., 2014*), (ii) analyzed SA currents from small diameter neurons (cell capacitance <30 pF) likely representing nociceptors (*Drew et al., 2004*; *Hu and Lewin, 2006*; *Poole et al., 2014*; *Woo et al., 2014*) and (iii) additionally normalized all obtained current data to the individual cell capacitance to obtain a value for current density (pA/pF). The latter analysis (data not shown) yielded similar results as the presented analysis based on current amplitudes for all DRG datasets in this manuscript. IA currents were randomly recorded across all neuron sizes.

Of note, current amplitudes and displacement thresholds cannot be compared among experimental datasets because of differential treatments (e.g. nucleofection, inhibitors) requiring measurements at different culture days in vitro (DIV), and due to the inherent variability in DRG cultures dependent on mouse cohorts used throughout the course of this study. For this reason each dataset consists of experiments and respective controls measured in parallel (i) in the same mouse cohort and (ii), where possible, on each experimental day. For each dataset (experimental versus control conditions) several coverslips from at least 4 (range: 4–18) independent cell cultures or 2–12 independent platings of HEK293 cells were used.

## Immunostaining and analysis

Immunostaining was carried out as described (*Narayanan et al., 2016*). Briefly, *Piezo2^GFP* mice (*Woo et al., 2014*) (ages 8–9 weeks) were euthanized with CO$_2$ or perfused with 4% PFA (Science Services). DRG neurons were isolated and cultured using the protocol described above or DRG were carefully dissected, collected in 4% PFA/1X PBS, and post-fixed for 30 min at 4°C. After overnight cryoprotection in 30% sucrose tissues were frozen in optimal cutting temperature medium, sectioned with a cryostat into 10 µm thick sections, mounted on SuperFrost Plus slides, and stored at −80°C. Frozen slides were thawed at room temperature for 30 min, washed thrice with 1X PBS, blocked for 30 min in 1X PBS containing 5% goat or donkey serum (Dianova, Germany) and 0.4% TritonX-100 (Roth), and incubated with primary antibodies (diluted in 0.1% TritonX-100% and 1% serum in 1X PBS), overnight at 4°C. The sections were then washed with 1X PBS and incubated for 2 hr at room temperature with secondary antibodies diluted 1:250 in 0.1% TritonX-100% and 1% serum in 1X

PBS. Sections were then washed six times with 1X PBS and mounted in SlowFade Gold antifade reagent with DAPI (Thermo Fisher Scientific).

For cultured DRG neurons, coverslips containing cells were washed with 1X PBS and fixed in 4% PFA for 10 min. Thereafter cells were washed with 1X PBS and blocked with blocking solution containing 5% serum and 0.4% TritonX-100. Cells were then incubated with primary antibodies (diluted in 1% serum and 0.1% TritonX-100), overnight at 4°C. Cells were then washed with 1X PBS and incubated with secondary antibodies (diluted in 1% serum and 0.1% TritonX-100) for 2 hr at room temperature followed by additional washes. Coverslips were mounted onto SuperFrost Plus slides using SlowFade Gold reagent with DAPI (Thermo Fisher Scientific).

WGA staining to mark membranes of cells was done as follows; before fixation, the cells were treated with WGA-555 (1:200) for 15 min at 37°C. Cells were then washed three times with medium and the standard immunostaining protocol was performed.

Imaging was done using a Zeiss Axio Observer Z1 inverted microscope or a Zeiss LSM 510 Meta Confocal microscope. All images were processed and analyzed by ImageJ, NIH (*Schindelin et al., 2015*).

For the analysis of Mtmr2 in cryo-frozen sections or primary cultures of DRGs from *Piezo2*[GFP] mouse, sections were stained with anti-GFP and anti-Mtmr2 antibodies. Positive cells were identified by setting the threshold as 'mean intensity +3*standard deviation' of randomly selected (~10) negative cells. The numbers of positive and negative cells were counted using the 'cell counter' plugin of ImageJ. Only for presentation purposes brightness, contrast and levels of images were adjusted in Adobe Photoshop. In all cases image adjustments were applied equally across the entire image and equally to controls.

For the analysis of membrane intensity of Piezo2-GFP upon Mtmr2 knockdown, the 'analyze particle tool' from ImageJ was used. The WGA staining was used to mark the region of interest around the cell membrane and for each ROI, the mean intensity (arbitrary units, AU) and area of positive signal was determined. The mean intensity of signal (arbitrary units, AU) was calculated by subtracting the threshold (defined as 'mean +3*standard deviation' of background) from the total mean intensity.

## Proximity ligation assay (PLA)

PLA was carried out as described (*Hanack et al., 2015*; *Narayanan et al., 2016*) with minor modifications. HEK293 cells or DRG neurons were plated on MatTek dishes coated with poly-D-lysine (and Laminin for DRG neuron cultures) and transfected with appropriate plasmids (details of plasmids are provided below). Cells were cultured for 48 hr before staining. Cells were washed with 1X PBS and fixed with 4% PFA for 10 min at room temperature. Thereafter cells were blocked in Duolink Blocking Solution (Sigma Aldrich, Germany) for 2 hr at room temperature. Cells were then incubated with primary antibodies (diluted in Duolink Antibody Diluent (Sigma Aldrich)), overnight at 4°C. Cells were washed with wash buffer A (0.01 M Tris, 0.15 M NaCl and 0.05% Tween 20, pH 7.4) and incubated with PLA probes (PLUS and MINUS probes were diluted 1:10 in Duolink antibody diluent (Sigma Aldrich)) for 1 hr at 37°C. Cells were washed again with wash buffer A and incubated with amplification mix (amplification stock 1:5 and polymerase 1:80 in water) for 100 min at 37°C. Cells were then washed with wash buffer B (0.2 M Tris, 0.1 M NaCl, pH7.5) and stored in 1X PBS before imaging. Secondary controls meant omitting all primary antibodies.

Plasmids used: pCMVSport6 *Piezo2-GST*-IRES-GFP (kind gift from Prof. Ardem Patapoutian); pCMV6-Entry *Mtmr2-myc-DDK* (Origene, #MR215223); pCMV6-Entry *Mtmr2C417S-mycDDK*, mutation as described by Berger and colleagues (*Berger et al., 2002*), was custom-generated using the Q5 Site-Directed Mutagenesis kit (New England BioLabs); pCMVSport6; pCDNA3.1-myc-His (Invitrogen, #V80020); pmaxGFPVector (Lonza).

The PLA was imaged using a Zeiss Axio Observer Z1 inverted microscope. The imaging settings were constant across all samples of the same experiments. Secondary controls were always imaged in parallel using the same settings. Image analysis was done using ImageJ. The background was determined as the 'mean intensity +3*standard deviation' of randomly chosen negative cells per field of view and averaged for all images within one condition. The highest background value was then used as threshold for the analysis. GFP positive cells (from Piezo2-GST-IRES-GFP expression or pmaxGFPVector) were chosen for each field of view and the PLA signal was analyzed for these cells, using the 'Analyze Particle' tool of ImageJ. The number of PLA puncta and total area of positive PLA

signal for each cell was measured. To account for variability in cell size, the total area of the cell was also measured and the PLA signal values were normalized to total cell area. Only for presentation purposes brightness, levels and contrast of images were adjusted in Adobe Photoshop. In all cases image adjustments were applied equally across the entire image and equally to controls. Experiments were performed on several coverslips of at least two independently transfected HEK293 and DRG cultures, respectively.

## Atomic force microscopy (AFM)

DRG neurons were nucleofected with AllStar Negative control siRNA (CTRL) or *Mtmr2* siRNA, as described above, and maintained in culture for 72 hr. Elasticity and tether force measurements were performed with an AFM (MFP-3D extended head, Asylum Research, Germany) mounted on an inverted microscope (IX71, Olympus, Germany) using contact mode with a triangular cantilever comprising a pyramidal tip (TR-400-PB, Olympus). During the measurements cells were maintained in growth media. The spring constant of the cantilever was determined using the built in thermal method (24–28 pN/nm). Indentation and retraction speed was kept constant at 5 µm/s, and force load of 200–1000 pN was used to measure the Young's modulus of the cells. The effective Young's modulus $E_{eff}$ was fitted with a modified Hertz model using a self-written IGOR macro (*Rehfeldt et al., 2007*). Tether forces were determined as the difference of pulling force before and after rupture of a tether from the AFM tip using a semi-automated step finding procedure (*Nawaz et al., 2015*).

## Bioinformatic identification of PIP2 binding regions in Piezo2

TRPML1 is known to bind $PI(3,5)P_2$ through a region in its N-terminus (*Dong et al., 2010*). This region of TRPML1 (NP_444407.1) was compared to mouse Piezo2 (NP_001034574.4) and mouse Piezo1 (NP_001032375.1) using NCBI protein Blast (National Library of Medicine (US), National Center for Biotechnology Information).

## Peptide-lipid binding assay

The protocol was adapted from (*Berger et al., 2002*) with minor modifications. In brief, PIP strips (Echelon Biosciences, Inc., Salt Lake City, USA) were washed once with PBS-T (1x PBS + 1% Tween) and blocked with 3% fat free BSA (bovine serum albumin; Sigma Aldrich) in 1X PBS for 1 hr at room temperature. The membranes were then incubated with 0.5 µg/mL peptide solution (peptide dissolved in 1% BSA) for 2 hr at room temperature. Experiments and controls were processed in parallel. Membranes were washed with PBS-T three times for 7 min each and then incubated with primary antibody at room temperature for 2 hr. Membranes were then washed with PBS-T and probed with secondary antibodies coupled with Alexa680 for 1 hr at room temperature. Imaging was done on the Odyssey Infrared System (LI-COR, Germany). Only for presentation purposes brightness, gradient levels and contrast of images were adjusted in Adobe Photoshop. In all cases image adjustments were applied equally across the entire image and equally to controls.

## Peptides

The following peptides were used in this study (all procured from GenScript, New Jersey, USA): Piezo2 (731-746)-FLAG tagged [EWWRKILKYFWMSVVIDYKDDDDKQNN]; Piezo2 3Q mutant (731-746)-FLAG tagged [EWWQQILQYFWMSVVIDYKDDDDKQNN]; Piezo1 (626-639)-FLAG tagged [TLWRKLLRVFWWLVDYKDDDDKqnn].

## Antibodies

The following antibodies were used in this study: 1:100 rabbit anti-Mtmr2 (Biotechne, Minneapolis, USA; #NBP1-33724); 1:500 chicken anti-GFP (Thermo Fisher Scientific, #A10262); 1:250 (Immunocytochemistry), 1:500 (PLA) rabbit anti-GST (Santa Cruz, Santa Cruz, USA; #sc-459); 1:100 (immunocytochemistry and immunoblotting), 1:750, 1:500 (PLA in HEK293 cells and DRG neurons respectively) mouse anti-myc (Santa Cruz, #sc-47694); 1:200 Rabbit anti-Piezo2 (Novus Biologicals, Germany; #NBP1-78624); 1:500 mouse anti-FLAG (Sigma Aldrich, #F1804), Secondary antibodies conjugated to Alexa Fluor 488, Alexa Fluor 546, Alexa Fluor 647, Alexa Fluor 680 (Thermo Fisher Scientific), Duolink in situ PLA probes 1:10 anti-rabbit MINUS, 1:10 anti-mouse PLUS.

## Statistics

Data was analyzed using GraphPad Prism 6.01 (San Diego, USA). All data are represented as mean ± SEM (standard error of mean) unless indicated otherwise. All replicates were biological. All statistical tests are two-sided unless indicated otherwise. In all panels: ns > 0.05; $*p \leq 0.05$; $**p \leq 0.01$; $***p \leq 0.001$; $****p \leq 0.0001$).

PLA data: Mann-Whitney test or Kruskal-Wallis test with Dunn's Multiple Comparison test.

Atomic force microscopy (AFM) data: Mann-Whitney test. qPCR: One sample t-test was used (values were compared to a theoretical mean of 1.00, that is mRNA expression in CTRL).

Immunostaining: For membrane Piezo2 expression and number of Piezo2-positive neurons upon Mtmr2 knockdown, the Mann-Whitney test was used.

Peptide-lipid binding assays: One-way ANOVA followed by Dunnett's or Holm-Sidak's multiple comparisons test was used as indicated.

Electrophysiology: For the analysis of stimulus-current curves, 2-way ANOVA with Holm-Sidak's multiple comparisons test was used. The P-value represents the results of 2-way ANOVA, testing the overall effect of the respective treatment. Results of the Holm-Sidak's multiple comparisons test are represented by p-values and indicated in each legend. Outlier analysis was carried out using the Grubb's test followed by testing whether the outlier value exceeded 'mean +3* standard deviation'. Outlier analysis was only performed on current values at maximal stimulation. Only if a value met both criteria (Grubb's outlier and >'mean +3*standard deviation') the cell was excluded from further analysis. Datasets, where a single outlier was removed: *Figure 1h*, *Figure 2b*, *Figure 4b*, *Figure 4d*, *Figure 4—figure supplement 1a,c*. For the analysis of the displacement threshold and inactivation time constant the Mann-Whitney test was used, unless more than two groups were compared, for which one-way ANOVA or Kruskal-Wallis test followed by Dunn's multiple comparisons test were used. For the analysis of mechanically activated (MA) current populations Chi-square test was used and data are represented as % of all analyzed cells.

## Acknowledgements

The authors would like to thank Ardem Patapoutian (HHMI, TSRI, La Jolla, USA) for kindly providing Piezo2$^{GFP}$ mice and *Piezo* plasmids, Walter Stühmer and Luis Pardo (both MPI for Experimental Medicine, Goettingen) for generously providing the electrophysiology equipment, and Kathrin Willig and Joris van Dort for preliminary experiments (both MPI for Experimental Medicine, Goettingen). Many thanks to Sergej Zeiter (MPI for Experimental Medicine, Goettingen) for excellent technical assistance. We are grateful to Ardem Patapoutian, Viktor Lukacs and Luis Pardo for thoughtful comments on the study and manuscript. This work was supported by the Emmy Noether-Program of the Deutsche Forschungsgemeinschaft (DFG) (SCHM 2533/2-1 to MS), the DFG Collaborative Research Center 889 (project A9 to MS; PN), the DFG Collaborative Research Center 937 (project A13 to FR; GK), a DFG research grant (DFG GO 2481/2-1 to DGV), the Max Planck Society and GGNB PhD fellowships (to PN and MH). The authors declare no competing financial interest.

## Additional information

### Funding

| Funder | Grant reference number | Author |
|---|---|---|
| Deutsche Forschungsgemeinschaft | SCHM 2533/2-1 | Manuela Schmidt |
| Max-Planck-Gesellschaft | Open-access funding | Meike Hütte<br>David Gomez-Varela<br>Manuela Schmidt |
| Göttinger Graduiertenschule für Neurowissenschaften, Biophysik und Molekulare Biowissenschaften | PhD fellowship | Pratibha Narayanan<br>Meike Hütte |
| Deutsche Forschungsgemeinschaft | CRC889 Project A9 | Manuela Schmidt |

| Deutsche Forschungsge-meinschaft | GO 2481/2-1 | David Gomez-Varela |
|---|---|---|
| Deutsche Forschungsge-meinschaft | CRC 937 Project A13 | Florian Rehfeldt |

The funders had no role in study design, data collection and interpretation, or the decision to submit the work for publication.

## Author contributions

Pratibha Narayanan, Formal analysis, Investigation, Visualization, Methodology, Writing—review and editing; Meike Hütte, Formal analysis, Investigation, Performed additional in vivo experiments; Galina Kudryasheva, Investigation, Performed AFM experiments; Francisco J Taberner, Formal analysis, Investigation, Performed preliminary skin-nerve-recordings; Stefan G Lechner, Formal analysis, Supervised and analyzed preliminary skin-nerve-recordings; Florian Rehfeldt, Formal analysis, Investigation, Writing—review and editing, Supervised and analyzed AFM experiments, Assisted with preparing the manuscript; David Gomez-Varela, Conceptualization, Formal analysis, Supervision, Funding acquisition, Methodology; Manuela Schmidt, Conceptualization, Formal analysis, Supervision, Funding acquisition, Validation, Visualization, Writing—original draft, Project administration, Writing—review and editing

## Author ORCIDs

Florian Rehfeldt http://orcid.org/0000-0001-9086-3835
Manuela Schmidt http://orcid.org/0000-0003-1972-3519

## Ethics

Animal experimentation: All experiments involving primary tissue isolated from mice were carried out in strict accordance with the recommendations of the institutional animal care and use committee (IACUC) of the Max Planck Institute of Experimental Medicine, Goettingen.

## Decision letter and Author response

Decision letter https://doi.org/10.7554/eLife.32346.018
Author response https://doi.org/10.7554/eLife.32346.019

# Additional files

## Supplementary files

• Supplementary file 1. Summary of properties of MA currents elicited under various conditions in HEK293 cells and DRG neurons The table shows the displacement threshold and inactivation time constant ($\tau$) values for all electrophysiological data presented in this study (please see Materials and methods for details on the calculation of each value). Values are represented as mean ± SEM and cell numbers are indicated by 'n'. Data were not significant (ns) unless otherwise mentioned. Please note: for SA-MA currents, it was not possible to fit the current traces of all cells with a mono or bi-exponential fit (please see Materials and methods for details), hence the cell numbers measured for the inactivation time constant ($\tau$) are lower than actual cell numbers measured and reported in *Figure 2*.
DOI: https://doi.org/10.7554/eLife.32346.015

• Transparent reporting form
DOI: https://doi.org/10.7554/eLife.32346.016

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
