## [Decision Letter]

Thank you for submitting your article "Mtmr2 and PI(3,5)P_2_ interdependently control Piezo2-mediated mechanotransduction in peripheral sensory neurons" for consideration by *eLife*. Your article has been reviewed by three peer reviewers, and the evaluation has been overseen by a Reviewing Editor and Richard Aldrich as the Senior Editor. The reviewers have opted to remain anonymous.

The reviewers have discussed the reviews with one another and the Reviewing Editor has drafted this decision to help you prepare a revised submission.

All three reviewers appreciated, very much, the concept of investigating membrane modulators of Piezos, which they consider as significant and timely. However, they raised strong concerns on your skin-nerve and behavior experiments.

Summary:

This manuscript reports a mechanism by which mechanically-activated Piezo2 ion channels are modulated by myotubularin-related protein-2 (Mtmr2) and PI(3,5)P_2_. Whereas Mtmr2 is a protein the same group recently identified as a novel interaction partner of Piezo2, the work here uncovers the functional significance. The data show that Mtmr2 through PI(3,5)P_2_ production regulates Piezo2 currents in HEK293 cells and DRG neurons, and present evidence for a binding site on Piezo2 for cellular PI(3,5)P_2_.

Essential revisions:

Overall, the area of work is of interest for the mechanotransduction field, as relatively little mechanistic information is known about how changes in the lipid membrane can affect somatosensation or how mechanotransduction channels are modulated. In addition to that, Mtmr2 has not been studied and the modulation of Piezo2 after injury is not well understood.

However, in its current form, although the manuscript is well written, it is difficult to conceptualize the mechanism that is proposed. The reviewers recognize that the lack of mechanistic framework is largely due to the inherent difficulty of the problem and the lack of tools. Moreover, strong concerns were raised about the skin-nerve and animal behavior experiments. For these reasons, the editor and the reviewers propose the following revisions to improve the message and strength of the manuscript.

1) Strong concerns about the skin-nerve and animal behavior experiments.

Using hypoosmotic stresses in vivo is not a classical approach to mimic tactile hypersensitivity rather than a more standard injury model that is typically used on the somatosensory field. Moreover, nerve recordings of the SA-Aβ fibers do not look like other published recordings of SA-Aβ fibers because their maximum stimulus response increases are not in the 5-20 mN range, which is typical for SA-Aβ fibers. There is concern that these recordings might have missed the ideal spot of the receptive field. The 200-300 mN range is too high for SA-Aβ fibers, and this atypical range is where effects on SA-Aβ could all be due to tissue damage (the abnormally high mechanical stimulus together with hypoosmotic stimuli). Even if these problems could be correctly addressed, since the basic mechanism is not clearly established, the in vivo results are difficult to interpret, as these effects can be due to a myriad of mechanisms other than Piezo2 regulation. For all of these reasons, we recommend deleting the skin nerve and animal behavioral assays aspect of your manuscript and to reinforce the mechanistic interactions as detailed below.

2) A key point would be to establish more precisely the specificity of Piezo2 for PI(3,5)P_2_ by performing new experiments with mutants.

Between mouse Piezo1 and Piezo2 the amino acid similarity within and proximal to this specific domain is extremely high:

Piezo1 LFLLCLTLFQVYYSLWRKLLKAFWWLVVAYTMLVLIAVYTFQFQDFP 661.

Piezo2 LFLFCVALYQVHYEWWRKILKYFWMSVVIYTMLVLIFIYTYQFENFP 768.

The questions to test are the following:

a) Can a Piezo1 peptide (RKLLKAFWWL) bind PI(3,5)P_2_?

b) Is a Piezo2 channel deficient in PI(3,5)P_2_ binding when mutated into the Piezo1 sequence (RKLLKAFWWL)?

c) Can Piezo1 be rendered into an Mtmr2-dependent channel by mutating it into the Piezo2 sequence (RKILKYFWMS)?

3) For the lipid binding data that are key to the paper, more quantification should be performed, since large batch to batch variability is a common issue and relying on a representative of n=3 is not very rigorous. It is important to show statistical summary. Moreover, it is very important to do same day controls with the Piezo1 and Piezo2 peptides.

[Editors' note: further revisions were requested prior to acceptance, as described below.]

Thank you for resubmitting your work entitled "Myotubularin related protein 2 and its phospholipid substrate PIP_2_ control Piezo2-mediated mechanotransduction in peripheral sensory neurons" for further consideration at *eLife*. Your revised article has been favorably evaluated by Richard Aldrich (Senior editor), a Reviewing editor, and two reviewers.

The manuscript has been improved but there are some remaining minor issues that need to be addressed before acceptance, as outlined below. Please add some lines of text to emphasise the two points.

1) The PLA "negative" controls show a clear signal, even though less than with Mtmr (Figure 1—figure supplement 1), confirming my suspicion about this technique. The authors show this in the supplemental figure but ignore it in the main text. I think it would be beneficial for the scientific community to draw attention to the propensity of this technique to produce artifacts especially in transfection systems. I think improper use of statistics is only a small part of the reproducibility crisis. Not being aware of the limitations of techniques is equally or even more important.

2) PI(3,5)P_2_ is thought to be mainly in intracellular membranes. The discussion acknowledges it but does so in the vaguest possible manner. I would probably be a little more direct and discuss this potential contradiction. It may trigger some future experiments to figure out what is going on.

---

## [Author Response]

Essential revisions:Overall, the area of work is of interest for the mechanotransduction field, as relatively little mechanistic information is known about how changes in the lipid membrane can affect somatosensation or how mechanotransduction channels are modulated. In addition to that, Mtmr2 has not been studied and the modulation of Piezo2 after injury is not well understood.However, in its current form, although the manuscript is well written, it is difficult to conceptualize the mechanism that is proposed. The reviewers recognize that the lack of mechanistic framework is largely due to the inherent difficulty of the problem and the lack of tools. Moreover, strong concerns were raised about the skin-nerve and animal behavior experiments. For these reasons, the editor and the reviewers propose the following revisions to improve the message and strength of the manuscript.1) Strong concerns about the skin-nerve and animal behavior experiments.Using hypoosmotic stresses in vivo is not a classical approach to mimic tactile hypersensitivity rather than a more standard injury model that is typically used on the somatosensory field. Moreover, nerve recordings of the SA-Aβ fibers do not look like other published recordings of SA-Aβ fibers because their maximum stimulus response increases are not in the 5-20 mN range, which is typical for SA-Aβ fibers. There is concern that these recordings might have missed the ideal spot of the receptive field. The 200-300 mN range is too high for SA-Aβ fibers, and this atypical range is where effects on SA-Aβ could all be due to tissue damage (the abnormally high mechanical stimulus together with hypoosmotic stimuli). Even if these problems could be correctly addressed, since the basic mechanism is not clearly established, the in vivo results are difficult to interpret, as these effects can be due to a myriad of mechanisms other than Piezo2 regulation. For all of these reasons, we recommend deleting the skin nerve and animal behavioral assays aspect of your manuscript and to reinforce the mechanistic interactions as detailed below.

We completely agree with the reviewers on the point that results of both experiments are difficult to interpret and might additionally reflect Piezo2-independent processes. We followed the reviewers´ suggestions and deleted the skin-nerve and animal behavioral assays from the revised manuscript.

2) A key point would be to establish more precisely the specificity of Piezo2 for PI(3,5)P_2_ by performing new experiments with mutants.Between mouse Piezo1 and Piezo2 the amino acid similarity within and proximal to this specific domain is extremely high:Piezo1 LFLLCLTLFQVYYSLWRKLLKAFWWLVVAYTMLVLIAVYTFQFQDFP 661.Piezo2 LFLFCVALYQVHYEWWRKILKYFWMSVVIYTMLVLIFIYTYQFENFP 768.The questions to test are the following:a) Can a Piezo1 peptide (RKLLKAFWWL) bind PI(3,5)P_2_?

Can a mouse Piezo1 peptide (mouse Piezo1 sequence: RKLLRVFWWL, please note that the sequence indicated by the reviewers represents the corresponding human Piezo1 region) bind PI(3,5)P_2_?

We tested this using lipid-peptide binding assays and did not observe any evidence that the corresponding region of mouse Piezo1 can bind any of the tested lipids including PI(3,5)P_2_. These experiments were always performed in parallel with additional experiments using the mouse Piezo2 peptide (serving as positive control) and the Piezo2 3Q mutant peptide. These data are included and discussed in the revised manuscript: Figure 5C,D; Subsection “Piezo2 harbors a PIP2 binding motif”; Discussion section.

b) Is a Piezo2 channel deficient in PI(3,5)P_2_ binding when mutated into the Piezo1 sequence (RKLLKAFWWL)?c) Can Piezo1 be rendered into an Mtmr2-dependent channel by mutating it into the Piezo2 sequence (RKILKYFWMS)?

These are excellent suggestions and we addressed them with additional experiments.

a) Can a mouse Piezo1 peptide (mouse Piezo1 sequence: RKLLRVFWWL, please note that the sequence indicated by the reviewers represents the corresponding human Piezo1 region) bind PI(3,5)P_2_?

We tested this using lipid-peptide binding assays and did not observe any evidence that the corresponding region of mouse Piezo1 can bind any of the tested lipids including PI(3,5)P_2_. These experiments were always performed in parallel with additional experiments using the mouse Piezo2 peptide (serving as positive control) and the Piezo2 3Q mutant peptide. These data are included and discussed in the revised manuscript: Figure 5C,D; Subsection “Piezo2 harbors a PIP2 binding motif”; Discussion section.

b) Is a Piezo2 channel deficient in PI(3,5)P_2_ binding when mutated into the mouse Piezo1 sequence (mouse Piezo1 sequence: RKLLRVFWWL, please note that the sequence indicated by the reviewers belongs to the corresponding human Piezo1 region)?

We generated this Piezo2 P1 mutant, which was functional upon heterologous expression in HEK293 cells. Remarkably, MA currents mediated by the Piezo2 P1 mutant were only slightly attenuated upon co-expression with Mtmr2 compared to mock-transfected controls (Figure 5D). Also, the displacement threshold and inactivation time constant (τ) of MA currents remained unchanged (Supplementary file 1). This result is in stark contrast to the pronounced Mtmr2-induced suppression of MA currents recorded from wildtype Piezo2 (Figure 1F).

Hence, these results suggest that the PI(3,5)P_2_ binding domain we identified in mouse Piezo2 indeed significantly contributes to the sensitivity of Piezo2 to Mtmr2-dependent changes in PI(3,5)P_2_ levels.

These data are included and discussed in the revised manuscript: Figure 5E; subsection “Piezo2 harbors a PIP2 binding motif”; Discussion section.

c) Can Piezo1 be rendered into an Mtmr2-dependent channel by mutating it into the Piezo2 sequence (RKILKYFWMS)?

We have generated this Piezo1 P2 mutant. Unfortunately, we could not obtain any functional recordings of RA-MA currents from this mutant despite intensive efforts, i.e. we tried to record from >40 cells from >7 independent transfections. To exclude any technical issues with recordings, we included the Piezo1 wildtype plasmid as a positive control (the same, which was used in Figure 1—figure supplement 2) and managed to successfully record RA-MA currents in parallel. To assess whether issues with the Piezo1 P2 mutant clone itself might be responsible for the lack of functionality we fully sequenced the Piezo1 P2 mutant, and additionally recloned the mutated region into a non-mutated backbone of pcDNA3.1 Piezo1-IRES-GFP. However, we did not obtain a functional Piezo1 P2 clone as assessed by electrophysiology.

Hence, despite all our efforts, we are not able to provide this dataset in our revised manuscript.

3) For the lipid binding data that are key to the paper, more quantification should be performed, since large batch to batch variability is a common issue and relying on a representative of n=3 is not very rigorous. It is important to show statistical summary. Moreover, it is very important to do same day controls with the Piezo1 and Piezo2 peptides.

This is certainly an important point. Therefore, we (i) added more experimental repetitions using both Piezo2 peptides, (ii) combined these experiments with lipid binding assays using Piezo1 in parallel on the same day (please see our reply to point 2 above), and (iii) added a statistical evaluation for all experiments. Of note, these additional experiments also uncovered weak, but statistically significant binding of the Piezo2 peptide to PI(3,4)P_2_. Taken together, these data suggest that not only positively charged amino acid residues, but also flanking amino acids in this Piezo2 PIP binding region contribute to PIP_2_ binding in a yet to be explored manner. Further, these in vitro binding studies substantiate our functional data on the specific link between Piezo2 and the Mtmr2 substrate PI(3,5)P_2_ by identifying a PI(3,5)P_2_ binding domain in Piezo2, but not in Piezo1.

These data are included and discussed in the revised manuscript: Figure 5B–D; Figure 5—figure supplement 1; Subsection “Piezo2 harbors a PIP2 binding motif”; Discussion section.

[Editors' note: further revisions were requested prior to acceptance, as described below.]

The manuscript has been improved but there are some remaining minor issues that need to be addressed before acceptance, as outlined below. Please add some lines of text to emphasise the two points.1) The PLA "negative" controls show a clear signal, even though less than with Mtmr (Figure 1—figure supplement 1), confirming my suspicion about this technique. The authors show this in the supplemental figure but ignore it in the main text. I think it would be beneficial for the scientific community to draw attention to the propensity of this technique to produce artifacts especially in transfection systems. I think improper use of statistics is only a small part of the reproducibility crisis. Not being aware of the limitations of techniques is equally or even more important.

The reviewers raise very important points. In the revised version of the manuscript we included a critical evaluation of PLA results in transfected HEK293 cells.

Results section: “For a more detailed subcellular analysis we used the proximity ligation assay (PLA). In this way we could show the close vicinity of Piezo2 and Mtmr2 in both, somata and neurites of cultured DRG neurons, and upon co-transfection in HEK293 cells (Figure 1A–D and Figure 1—figure supplement 1C,D). It is important to note here that the PLA technique is prone to high background upon heterologous expression as shown by our additional control experiments in HEK293 cells (Figure 1—figure supplement 1D). In these we co-overexpressed Piezo2 with TRPA1 and Vti1b, respectively. Both of these controls exhibited clear PLA signal (likely attributable to massive overexpression upon transfection), though less than co-overexpression with Mtmr2 (Figure 1—figure supplement 1D).”

2) PI(3,5)P_2_ is thought to be mainly in intracellular membranes. The discussion acknowledges it but does so in the vaguest possible manner. I would probably be a little more direct and discuss this potential contradiction. It may trigger some future experiments to figure out what is going on.

We now included a more detailed and directed discussion on the possibility that Piezo2 might be regulated in intracellular membranes. Furthermore, we specifically mention this possibility in the legend of Figure 6.

Discussion section: “It remains to be seen whether these membrane compartments are confined to the plasma membrane and/or to intracellular membranes such as endo-lysosomes, which are known to contain the majority of PI(3,5)P_2_ (Di Paolo and De Camilli, 2006; Dong et al., 2010; Zolov et al., 2012). Along these lines it is worth mentioning that AMPA receptor abundance at hippocampal synapses has been shown to be regulated by PI(3,5)P_2_-controlled cycling through early and late endosomes (McCartney et al., 2014). While unknown so far, localization of Piezo2 to intracellular membranes would not be unexpected (Coste et al., 2010) since its family member Piezo1 has originally been described to reside in the endoplasmic reticulum (McHugh et al., 2010). Thus, exploring endocytosis and intracellular trafficking of Piezo2 may offer novel insights into its regulation by the intracellular membrane pool of PI(3,5)P_2_.

Figure legend 6: “This would provide an active mechanism to inhibit Piezo2 RA-MA currents in membrane compartments – may they be at the plasma membrane or intracellular membranes.”